# Non-coding RNAs participate in the regulatory network of CLDN4 via ceRNA mediated miRNA evasion

Yong-xi Song[1], Jing-xu Sun[1], Jun-hua Zhao[1], Yu-chong Yang[1], Jin-xin Shi[1], Zhong-hua Wu[1], Xiao-wan Chen[1], Peng Gao[1], Zhi-feng Miao[1,2] & Zhen-ning Wang[1]

Thousands of genes have been well demonstrated to play important roles in cancer progression. As genes do not function in isolation, they can be grouped into "networks" based on their interactions. In this study, we discover a network regulating Claudin-4 in gastric cancer. We observe that Claudin-4 is up-regulated in gastric cancer and is associated with poor prognosis. Claudin-4 reinforce proliferation, invasion, and EMT in AGS, HGC-27, and SGC-7901 cells, which could be reversed by miR-596 and miR-3620-3p. In addition, lncRNA-KRTAP5-AS1 and lncRNA-TUBB2A could act as competing endogenous RNAs to affect the function of Claudin-4. Our results suggest that non-coding RNAs play important roles in the regulatory network of Claudin-4. As such, non-coding RNAs should be considered as potential biomarkers and therapeutic targets against gastric cancer.

[1] Department of Surgical Oncology and General Surgery, The First Affiliated Hospital of China Medical University, 155 North Nanjing Street, Heping District, Shenyang City 110001, China. [2] Division of Gastroenterology, Department of Medicine, Washington University School of Medicine, St Louis, MO 63110, USA. Yong-xi Song, Jing-xu Sun and Jun-hua Zhao contributed equally to this work. Correspondence and requests for materials should be addressed to Z.-n.W. (email: josieon826@sina.cn)

Gastric cancer (GC) is one of the most common cancers, with a heavy mortality rate all over the world[1]. Metastasis, an end consequence of numerous complex processes, presents a major challenge in clinical practice and accounts for a major source of mortality and recurrence in GC[2]. During the complex process of metastasis, primary cancer cells undergo a sequential series of events including local dissemination, intravasation into the vascular system, survival in the circulatory system, extravasation out of the vascular system, and regrowth at distant sites[3–5]. Up to now, many molecular mechanisms of metastasis have been investigated, but the role of potential networks between mRNA and non-coding RNAs (ncRNAs) has not been fully elucidated.

Recently, studies have started to characterize the regulatory effects that ncRNAs may have on GC[6–8]. These ncRNAs are closely associated with the occurrence, development, invasion, and metastasis of tumors, as well as drug resistance[9–13]. Among these ncRNAs, microRNAs (miRNAs) and long non-coding RNAs (lncRNAs) have appealed to a large group of researchers and become a main focus of attention. Increasing evidence has uncovered the indispensable function of miRNAs in post-transcriptional regulation of oncogenes and tumor suppressor genes, thus modulating the biological behaviors of tumor cells such as invasion, metastasis, proliferation, and apoptosis[14, 15]. LncRNAs are now known to have many functions, acting as scaffolds or guides to regulate interactions between protein and genes, as decoys to bind proteins or miRNAs, and as enhancers to modulate transcription of their targets after being transcribed from enhancer regions or their neighboring loci[16–22]. Moreover, increasing studies have indicated that some discrepantly expressed lncRNAs possess significant regulatory effects on carcinogenesis and the development of cancer, demonstrating their potential roles in both oncogenic and tumor-suppressive pathways[23–27]. Intriguingly, some recent studies report a brand-new lncRNAs regulatory circuitry in which lncRNAs may function as competing endogenous RNAs (ceRNAs) and crosstalk with mRNAs by competitively binding their common miRNAs[23, 28–30].

In the present study, six pairs of gastric cancer tissues and non-tumorous adjacent tissues were analyzed using microarray, and abnormally expressed mRNAs, miRNAs, and lncRNAs were selected for deep analysis. Following bioinformatic analyses, the network of claudin-4 (CLDN4), which is involved in metastasis of GC, captured our attention. The claudin family is well known for its pivotal role in the constitution and maintenance of tight junctions[31]. CLDN4, a critical member of the claudin family, has been observed to alter expression patterns in various types of carcinomas including gastric cancer[32], pancreatic cancer[33], and ovarian cancer[34]. In addition, we have previously demonstrated aberrant expression of CLDN4 in GC and precursor lesions[35]. Using meta-analysis, we have also found that CLDN4 expression is associated with increasing pT category, tumor size, and lymph node metastasis in patients with GC[36]. Simultaneously, accumulating evidence confirms that aberrant expression of CLDN4 may result in an intense tendency towards metastasis of cancers, mainly because CLDN4 can enhance the invasion capacity of cancer cells and promote epithelial-mesenchymal transition (EMT)[37, 38]. Overexpression of CLDN4 is positively associated with the expression of metalloproteinase-2 (MMP-2) and metalloproteinase-9 (MMP-9), both of which exhibit the ability to degrade components of the extracellular matrix and eventually reinforce the invasive capacity and motility of cancer cells[38–40]. Particularly, due to the critical role of CLDN4 in the formation of tight junctions, disruption and dysfunction of tight junctions originating from the aberrant expression of CLDN4 may decrease the stability of cell-to-cell adhesions and thus facilitate the detachment and metastasis of cancer cells.

Although there have been several studies focusing on investigating the expression profile and function of CLDN4 in different cancers, the upstream regulatory mechanism of CLDN4 has rarely been explored until now. Neither miRNAs nor lncRNAs have been reported to participate in the direct regulation of CLDN4. In our current work, we have discovered the existence of several miRNAs and lncRNAs which may cause abnormal expression of CLDN4. Furthermore, on the basis of microarray and experimental analyses, we propose a regulatory network in which CLDN4 is regulated by these ncRNAs in a ceRNA-mediated miRNA evasion, thus contributing to the metastasis and progression of GC.

## Results

**CLDN4 is identified as a target of miR-596 and miR-3620-3p.** In an attempt to identify the regulatory networks of mRNA and ncRNAs in GC, six pairs of GC tissues and non-tumorous adjacent tissues were analyzed via microarray using the Human LncRNA+mRNA Array v3.0 together with the miRCURY LNA™ microRNA Array. These six patients consisted of four males and two females, with an average age of 66.84 years. Detailed characteristics and values for each individual patient are shown in Supplementary Table 1 and Supplementary Data 1–3. In total, 235 miRNAs were found to be differentially expressed between GC and non-tumorous adjacent tissues (Supplementary Data 4), including 173 that were up-regulated and 62 that were down-regulated more than two fold. In total 4329 lncRNAs were found to be significantly differentially expressed (Supplementary Data 5), of which 1974 were up-regulated and 2355 down-regulated. In total 3369 differentially expressed mRNAs were also identified, with 1816 up-regulated and 1553 down-regulated more than two fold (Supplementary Data 6). Hierarchical clustering was applied to show expression patterns of miRNA, lncRNA, and mRNA (Fig. 1a). These mRNA microarray results were then intersected with predicted target genes of the differentially expressed miRNAs (Supplementary Data 7). Moreover, GO analysis (Supplementary Fig. 1a, b) and pathway analysis (Supplementary Fig. 1c, Supplementary Data 8) were also applied to analyze the differentially expressed mRNAs. GO analysis indicated the most significant Biological Processes, Cellular Components, and Molecular Function. Pathway analysis indicated that the most significant pathways consisted of *Staphylcoccus aureus* infection, cell cycle, chemical carcinogenesis, and so on. According to the pathway analysis results, we selected four classical pathways related to cancer development: "Cell adhesion", "Pathway in cancer", "Tight junction", and "Cell cycle". Combined with the lncRNA and miRNA microarray results, an mRNA-ceRNA analysis was performed on choosing classical genes within these four pathways (Fig. 1b). Details regarding the process of pathway analysis, mRNA-ceRNA analysis, and selection of key genes are listed in Supplementary Methods.

In the mRNA-ceRNA analysis results for the "Tight junction" pathway, CLDN4, which is closely related to cancer development[41, 42], drew our attention. Validations were performed to verify the relationship between CLDN4 and its correlative miRNAs. Initially, TargetScan (Release 7.1) results showed that CLDN4 contained predicted miR-596, miR-3620-3p, and miR-4292 targeting sites (Fig. 1c). We verified that CLDN4 expression could be negatively regulated by miR-596, miR-3620-3p, and miR-4292 at both the transcriptional (Fig. 1d) and translational (Fig. 1e) levels, as measured by real-time PCR and western blotting, respectively. To test the effects of these miRNAs on gene expression, we transfected the luciferase reporter plasmid psiCHECK2-CLDN4 into GC cells.

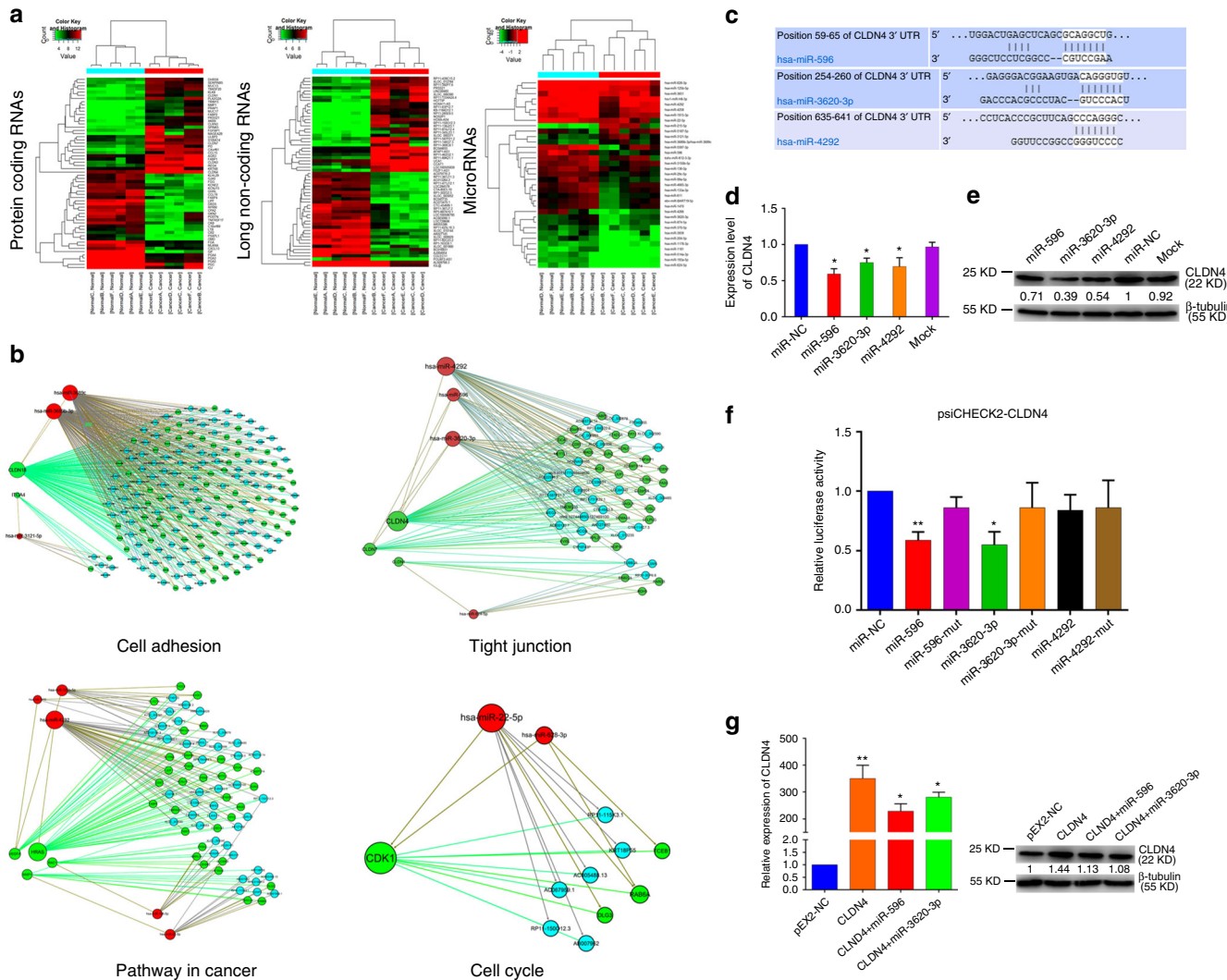

**Fig. 1** CLDN4 is identified as a target of miR-596 and miR-3620-3p. **a** Hierarchical clustering analysis of mRNAs, lncRNAs, and miRNAs that were differentially expressed between GC tissues and non-tumorous adjacent tissues (>2.0-fold; $P < 0.05$; filtered to show the top 30 up-regulated or down-regulated results for mRNAs and lncRNAs). Expression values are represented in shades of *red* and *green*, indicating expression above and below the median expression value across all tissues, respectively. **b** The mRNA-lncRNA -miRNA networks in the GC. The networks include cell adhesion pathway, pathway in cancer, tight junction pathway, and cell cycle pathway. Genes colored in *green* are protein-coding RNAs associated with GC. Genes colored in *blue* are lncRNAs and genes colored in *red* are miRNAs associated with GC. **c** Predicted binding sites for miR-596, miR-3620-3p, and miR-4292 on the CLDN4 transcript. The white nucleotides are the seed sequences of miRNAs. **d** Real-time PCR analysis of CLDN4 expression in GC cells treated with mimics of miR-596, miR-3620-3p, miR-4292, and negative control. **e** GC cell line SGC-7901 was transfected with the mimics of miR-596, miR-3620-3p, miR-4292, and negative control. Reduced CLDN4 expression was shown by western blotting analysis and normalized to β-tubulin. **f** Luciferase activities were measured in GC cells co-transfected with luciferase reporter containing CLDN4 and the mimics of miR-596, miR-3620-3p, miR-4292, or mutant. Data are presented as the relative ratio of renilla luciferase activity and firefly luciferase activity. **g** The relative expressions of CLDN4 were determined by real-time PCR and western blotting. Data are shown as mean ± s.d., $n = 3$. The data statistical significance is assessed by Student's *t*-test. *$P < 0.05$, **$P < 0.01$

Overexpression of miR-596 and miR-3620-3p, but not miR-4292, miR-596-mut, miR-3620-3p-mut, miR-4292-mut, or miRNA negative control (miR-NC), decreased the luciferase activity of psiCHECK2-CLDN4 (Fig. 1f). In summary, according to microarray results and bioinformatic analysis, we selected four pathway networks and identified CLDN4 as a target of interest. Through real-time PCR, western blotting, and luciferase assays, we verified the network related to CLDN4 and proved that CLDN4 is a target gene of miR-596 and miR-3620-3p.

**The biological function of CLDN₄ in vitro**. To test the biological function of CLDN4 and further verify its association with miR-

596 and miR-3620-3p, we stably overexpressed CLDN4 in SGC-7901, AGS, and HGC-27 cells (Supplementary Fig. 2a). This CLDN4 up-regulation caused by stable transfection could be overcome at both the transcriptional and translational levels by ectopic overexpression of miR-596 or miR-3620-3p (Fig. 1g). By performing Cell Counting Kit-8 (CCK-8) assays, we found that CLDN4 overexpression could significantly increase the proliferative capacity of SGC-7901, AGS, and HGC-27 cells compared with parallel stable cell lines containing empty vector pEX-2 (pEX2-NC cells). This increase could be eliminated when miR-596 or miR-3620-3p, but not miR-596-mut or miR-3620-3p-mut, were transfected. To examine the effect of CLDN4 on cell invasion ability, we performed transwell experiments and scrape

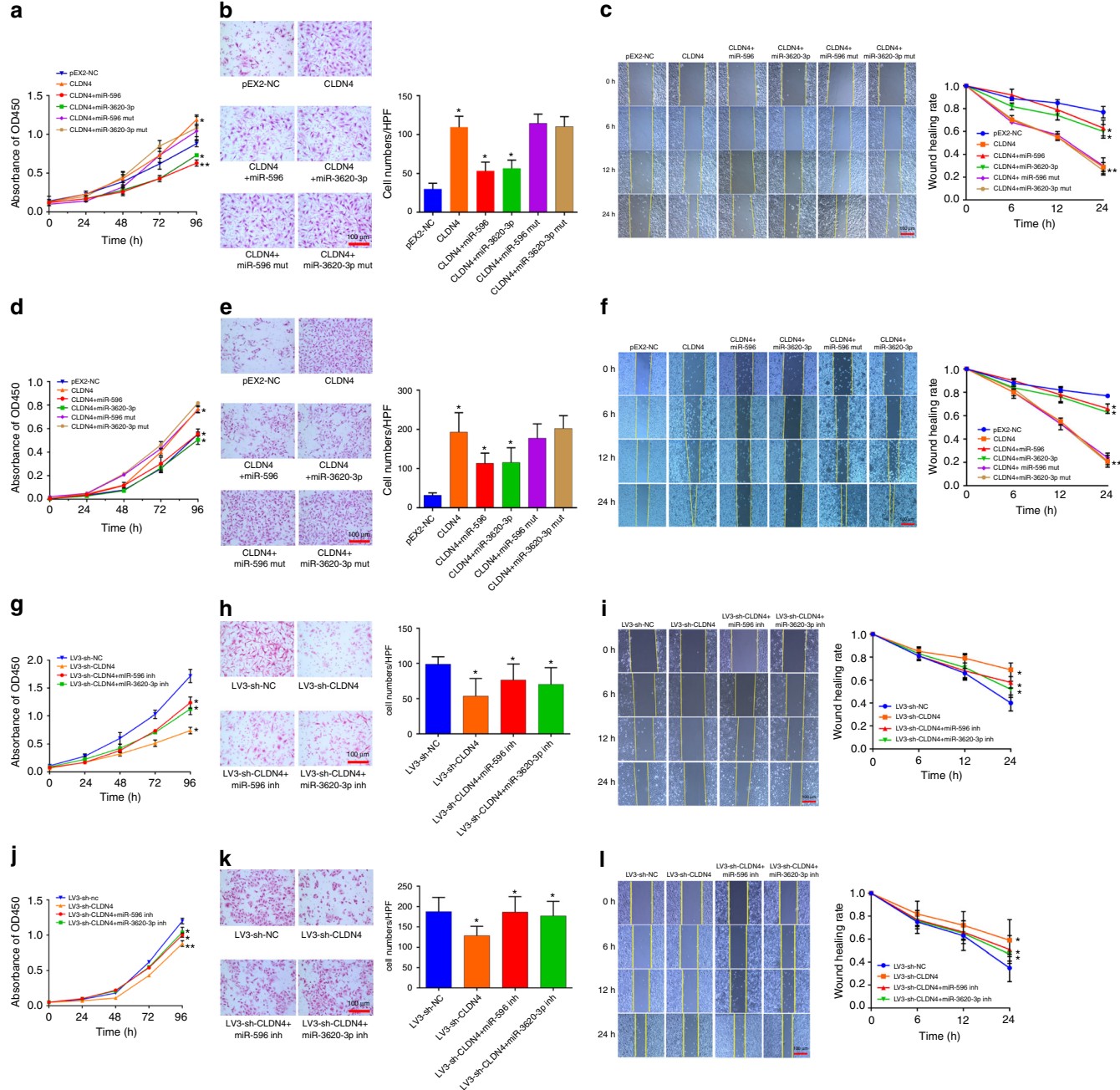

**Fig. 2** CLDN4 reinforces the proliferative and invasive capacity of GC cells in vitro. **a** Cell proliferation was assessed daily for 4 days using the Cell Counting Kit-8 (CCK-8) assay in CLDN4 overexpressing SGC-7901 cells. **b** Transwell assays were used to evaluate the involvement of CLDN4 for invasion in CLDN4 overexpressing SGC-7901 cells. **c** Scrape motility assays were monitored at 0, 6, 12, and 24 h in CLDN4 overexpressing SGC-7901 cells. **d** Cell proliferation assessed in CLDN4 overexpressing AGS cells. **e** Transwell assays assessed in CLDN4 overexpressing AGS cells. **f** Scrape motility assays in CLDN4 overexpressing AGS cells. **g** Cell proliferation assessed in CLDN4 knockdown SGC-7901 cells. **h** Transwell assays assessed in CLDN4 knockdown SGC-7901 cells. **i** Scrape motility assays in CLDN4 knockdown SGC-7901 cells. **j** Cell proliferation assessed in CLDN4 knockdown AGS cells. **k** Transwell assays assessed in CLDN4 knockdown AGS cells. **l** Scrape motility assays in CLDN4 knockdown AGS cells. In **b**, **e**, **h**, and **k**, cells were incubated for 24 h, and counted under the microscope. Original magnification ×200. *Scale bars* = 100 μm. Data are shown as mean ± s.d., $n = 3$. The data statistical significance is assessed by Student's *t*-test. *$P < 0.05$, **$P < 0.01$

motility assays in vitro. These experiments showed that CLDN4 overexpression could significantly increase the invasion ability of GC cells, compared with pEX2-NC cells, and this increase could again be partially abolished when miR-596 or miR-3620-3p were transfected (Fig. 2a–f, Supplementary Fig. 2b, c). To further support these results, we stably knocked down endogenous CLDN4 in SGC-7901, AGS, and HGC-27 cells using lentiviral shRNA (LV3-sh-CLDN4 cells) (Supplementary Fig. 2d). The

CCK-8 assays showed that the proliferation of LV3-sh-CLDN4 cells was significantly slower than parallel stable cell lines transfected with scrambled shRNA (LV3-sh-NC cells). LV3-sh-CLDN4 cells also showed attenuated invasion ability compared with LV3-sh-NC cells. The reduction in proliferative capacity and invasion ability caused by knocking down CLDN4 could be largely rescued by inhibition of miR-596 or miR-3620-3p (Fig. 2g–l, Supplementary Fig. 2e, f).

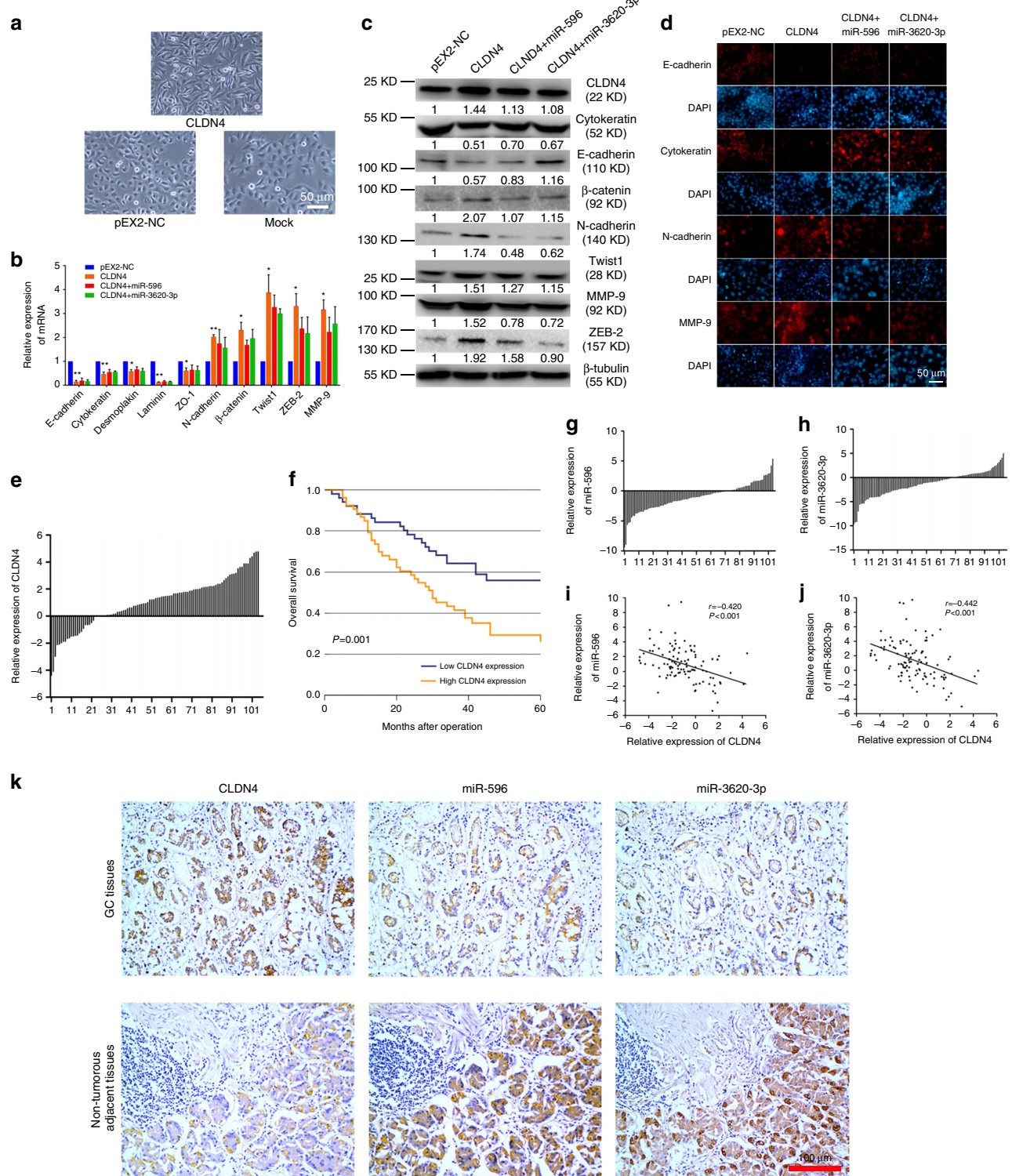

**Fig. 3** CLDN4 induces EMT in vitro. **a** Phase-contrast micrographs of CLDN4 overexpressing cells, pEX2-NC cells and SGC-7901 cells. *Scale bars* = 50 μm. **b**, **c** The transcriptional and translational levels of EMT related markers. The real-time PCR and western blotting were performed at 48 h after the CLDN4 overexpressing cells treated with mimics of miR-596 or miR-3620-3p. **d** Immunofluorescence microscopy of the localization and expression of EMT and invasion markers in CLDN4 overexpressing cells. *Scale bars* = 50 μm. **e** The relative expression levels of CLDN4 in human GC tissues compared with their matched non-tumorous adjacent tissues. **f** Kaplan–Meier analysis of the correlation between CLDN4 expression levels and overall survival. **g**, **h** The relative expression levels of miR-596 and miR-3620-3p in human GC tissues compared with their matched non-tumorous adjacent tissues. **i**, **j** The correlation between CLDN4 transcriptional levels and miR-596 or miR-3620-3p transcriptional levels were measured in the same set of patients by Spearman correlation analysis. **k** Representative images of CLDN4, miR-596 and miR-3620-3p expression from GC tissues and non-tumorous adjacent tissues by ISH assays. Original magnification × 200. *Scale bars* = 100 μm. Data are shown as mean ± s.d., *n* = 3. The data statistical significance is assessed by Student's *t*-test and Wilcoxon's Sign Rank Test is used to evaluate the differential expression between GC tissues and their matched non-tumorous adjacent tissues. *$P < 0.05$, **$P < 0.01$

Having found that CLDN4 could regulate cell proliferation, we analyzed differences in cell-cycle distribution and apoptosis following CLDN4 overexpression to further investigate its mechanism. We noted that CLDN4 had no obvious influence on cell-cycle distribution (Supplementary Fig. 3a) or apoptosis (Supplementary Fig. 3b). To our surprise, miR-596 or miR-3620-3p could promote apoptosis (Supplementary Fig. 3b), suggesting that other target genes of the miRNAs may contribute to this interesting result. Afterwards, to evaluate whether the effects of these two miRNAs on cell proliferation and invasion were mainly dependent on CLDN4, we transfected the two miRNAs into CLDN4 knockdown cells. The two miRNAs had little effect on cell proliferation and metastasis when CLDN4 was knocked down (Supplementary Fig. 3c, d).

While performing the above experiments, we noticed that CLDN4 overexpression induced mesenchymal-like morphological features in GC cells (Fig. 3a). We subsequently studied the correlations between CLDN4 and EMT-related genes at both the transcriptional and translational levels to further investigate the mechanism of how CLDN4 may regulate cell invasion (Supplementary Fig. 4a, b). We analyzed epithelial markers (E-cadherin, syndecan 1, mucin-1, Cytokeratin, desmoplakin, laminin, and ZO-1), mesenchymal markers (N-cadherin, β-catenin, fibronectin 1, α-SMA, Twist1, FOXC2, FSP-1, SNAI1, slug, vimentin, ZEB-1, and ZEB-2), and invasion related markers (MMP2, MMP9) via real-time PCR and western blotting. We saw that overexpression of CLDN4, compared with pEX2-NC cells, could reduce E-cadherin and Cytokeratin expression and enhance expression of N-cadherin, β-catenin, Twist1, ZEB-2, and MMP9. As expected, miR-596 and miR-3620-3p could partially abolish these effects (Fig. 3b, c). Validations performed in AGS and HGC-27 cells showed similar results to those in SGC-7901 cells supporting that CLDN4 could promote EMT across cell lines (Supplementary Fig. 4c, d). Immunofluorescence staining was used to further confirm these observations. We found that CLDN4 induced loss of E-cadherin and Cytokeratin expression and increased N-cadherin and MMP-9, all of which were partially abolished by miR-596 and miR-3620-3p transfection (Fig. 3d). In summary, through functional experiments, real-time PCR, western blotting, and immunofluorescence staining, we revealed functions of CLDN4 in promoting cell proliferation, invasion, and EMT. Simultaneously, we revealed that miR-596 and miR-3620-3p could partially abolish these functions.

**The expression of CLDN4 and miRNAs in GC tissues**. To further investigate the role of CLDN4 in human GC, we performed deep validations in 104 pairs of GC tissues and non-tumorous adjacent tissues. With 76.92% of patients showing higher expression in cancer tissue, the transcriptional level of CLDN4 was significantly higher in the GC tissues compared with matched non-tumorous adjacent tissues via Wilcoxon's Sign Rank Test ($P < 0.001$, Fig. 3e, Supplementary Fig. 4e). Subsequently, we examined whether CLDN4 expression was related to the prognosis of GC after gastrectomy. Using the Kaplan–Meier analysis with log-rank test for these 104 patients, we found that higher CLDN4 expression level was significantly correlated with decreased overall survival ($P = 0.001$, Fig. 3f). Furthermore, the result of Cox multivariate analysis revealed that higher CLDN4 expression was an independent predictor for poor prognosis in GC (HR = 2.634, 95% CI = 1.536–4.517, $P < 0.001$, Supplementary Table 2). As CLDN4 has been proven to be a target gene of miR-596 and miR-3620-3p, we also examined the expression of these miRNAs in the same tissues. The expression levels of miR-596 and miR-3620-3p were both significantly lower in the GC tissues ($P < 0.001$ for both, Supplementary Fig. 4e). In addition

70.19% and 66.35% of patients showed lower miR-596 expression and miR-3620-3p expression in GC tissues compared with matched non-tumorous adjacent tissues, respectively (Fig. 3g, h). Not surprisingly, the significant negative correlations between CLDN4 expression and miR-596, miR-3620-3p expression were confirmed by the Spearman correlation coefficients ($P < 0.001$ for both, Fig. 3i, j, Supplementary Fig. 5). These data support that high CLDN4 expression in GC tissues is associated with poorer survival and negatively associated with expression of miR-596 and miR-3620-3p. In situ hybridization (ISH) results in 20 cases of GC showed that CLDN4, miR-596, and miR-3620-3p were expressed by epithelial and carcinoma cells and mainly localized in the cytoplasm. Furthermore, the expression patterns detected by ISH were consistent with the real-time PCR results (Fig. 3k).

**LncRNAs can crosstalk with miRNAs through direct binding**. To identify lncRNAs which may interact with miR-596 or miR-3620-3p and serve as ceRNAs, we performed experiments focusing on the previously mentioned "Tight junction" network (Fig. 1b). Among 14 candidate lncRNAs, lncRNA-KRTAP5-AS1 presented the most obvious down-regulation in response to overexpression of miR-596. Simultaneously, expression of lncRNA-TUBB2A and lncRNA-KRTAP5-AS1 decreased the most following miR-3620-3p overexpression (Fig. 4a). To test the specificity of these regulatory interactions, we performed RNA-sequencing on six gastric cancer cell lines (SGC-7901 cells overexpressing miR-596, overexpressing miR-3620-3p, miRNA negative control, overexpressing lncRNA-KRTAP5-AS1, overexpressing lncRNA-TUBB2A, or lncRNA negative control). In cells overexpressing miR-596, the expression of CLDN4 (fold-change of FPKM = 0.125) and lncRNA-KRTAP5-AS1 (fold-change of FPKM = 0.760) were down-regulated. In addition, the expression of CLDN4 (fold-change of FPKM = 0.543), lncRNA-TUBB2A (fold-change of FPKM = 0.854), and lncRNA-KRTAP5-AS1 (fold-change of FPKM = 0.280) were all down-regulated in cells overexpressing miR-3620-3p compared with the control group. Interestingly, increased CLDN4 expression was found in cells overexpressing lncRNA-KRTAP5-AS1 (fold-change of FPKM = 14.168) or lncRNA-TUBB2A (fold-change of FPKM = 12.051) (Supplementary Data 9).

LncRNA-KRTAP5-AS1, located on chromosome 11 and with a length of 2554 nucleotides, has five predicted miR-3620-3p targeting sites and five predicted miR-596 targeting sites. LncRNA-TUBB2A, located on chromosome 6 and with a length of 1052 nucleotides, has five predicted miR-3620-3p targeting sites (Supplementary Figs. 6, 7). We found that lncRNA-KRTAP5-AS1 and lncRNA-TUBB2A presented the same cytoplasmic localization as CLDN4 and the miRNAs through ISH. We then transfected the luciferase reporter plasmids psiCHECK2-lncRNA-TUBB2A and psiCHECK2-lncRNA-KRTAP5-AS1 into GC cells to test for potential effects that the miRNAs may have on the expression of the lncRNAs. Overexpression of miR-596 and miR-3620-3p, but not miR-596-mut, miR-3620-3p-mut, or miR-NC, decreased the luciferase activity of psiCHECK2-KRTAP5-AS1 (Fig. 4b). The luciferase activity of psiCHECK2-TUBB2A could be decreased by miR-3620-3p, but not miR-3620-3p-mut or miR-NC (Fig. 4c). Afterwards, we performed RNA immunoprecipitation (RIP) experiments to further investigate the potential direct binding between lncRNA-KRTAP5-AS1, lncRNA-TUBB2A, and the miRNAs. To construct plasmids that could produce lncRNAs identified by the MS2 protein, we subcloned an MS2-12X fragment into pcDNA3.1, pcDNA3.1- KRTAP5-AS1, pcDNA3.1-TUBB2A, pcDNA3.1-KRTAP5-AS1-mut, and pcDNA3.1-TUBB2A-mut plasmids. We also constructed a GFP and MS2 gene fusion

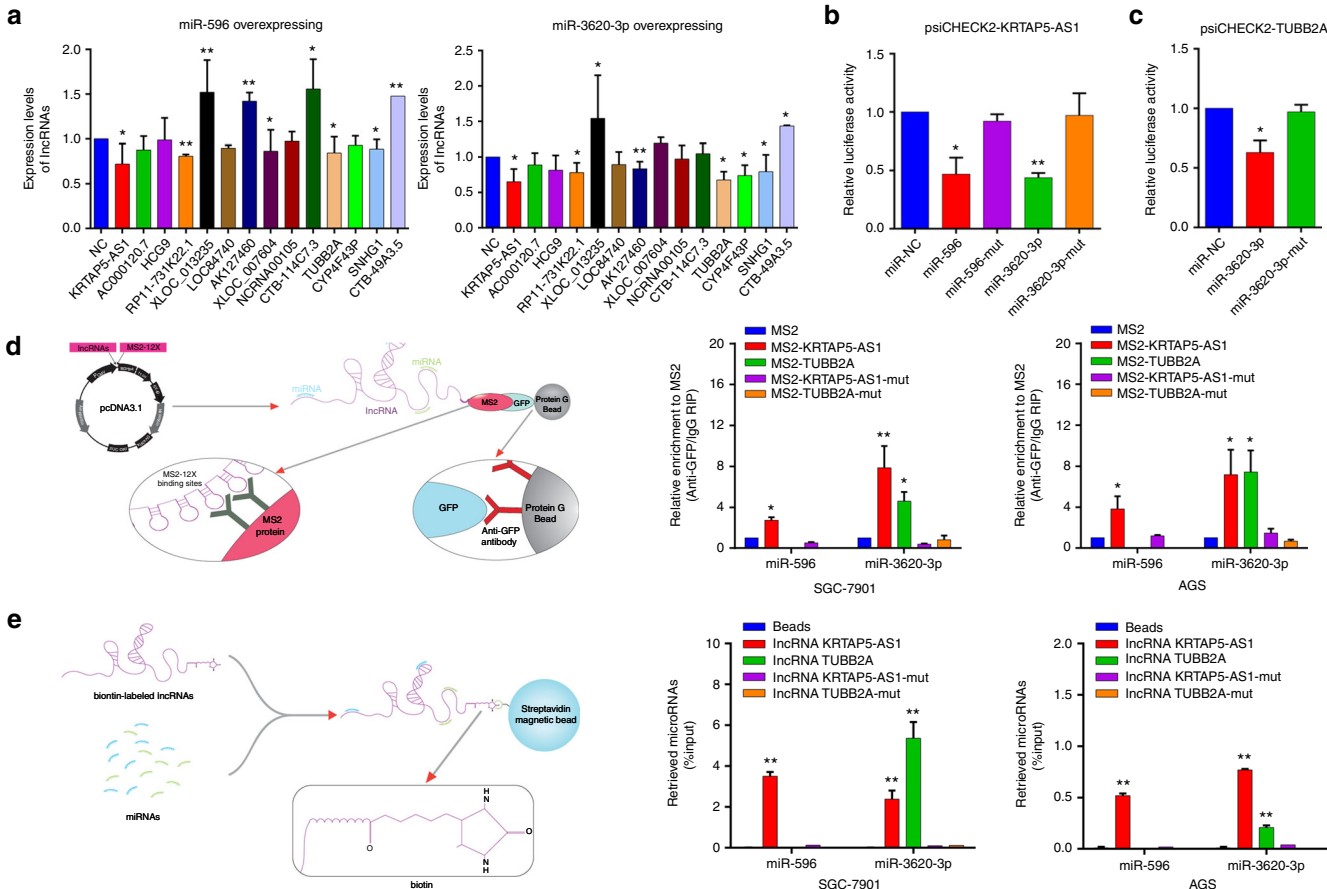

**Fig. 4** LncRNAs can crosstalk with miRNAs through direct binding. **a** Relative expression of lncRNAs in GC cells treated with mimics of miR-596 or miR-3620-3p were measured by real-time PCR. **b** Luciferase activity in GC cells co-transfected with luciferase reporter containing lncRNA-KRTAP5-AS1 and the mimics of miR-596, miR-3620-3p or mutant. Data are presented as the relative ratio of renilla luciferase activity and firefly luciferase activity. **c** Luciferase activity in GC cells co-transfected with luciferase reporter containing lncRNA-TUBB2A and the mimics of miR-3620-3p or mutant. Data are presented as the relative ratio of renilla luciferase activity and firefly luciferase activity. **d** The *schematic diagram* and real-time PCR results of the MS2-RIP method used to identify the binding between lncRNAs and miRNAs in both SGC-7901 and AGS cells. **e** The *schematic diagram* of the RNA pull down method used to identify the binding between lncRNAs and miRNAs in both SGC-7901 and AGS cells. GC cell lysates were incubated with biotin-labeled lncRNA-KRTAP5-AS1, lncRNA-TUBB2A, lncRNA-KRTAP5-AS1-mut, and lncRNA-TUBB2A-mut. MiRNA real-time PCR was performed after pull down process. Data are shown as mean ± s.d., *n* = 3. The data statistical significance is assessed by Student's t-test. *P < 0.05, **P < 0.01

expression vector to produce a GFP-MS2 fusion protein which could specifically bind the MS2-12X fragment and be identified using an anti-GFP antibody. Thus, miRNAs which interact with lncRNA-KRTAP5-AS1 or lncRNA-TUBB2A could be pulled down by the GFP-MS2-lncRNA compounds and analyzed via real-time PCR. We observed that the lncRNA-KRTAP5-AS1 RIP in SGC-7901 and AGS cells was significantly enriched for both miR-596 and miR-3620-3p compared to pcDNA3.1-KRTAP5-AS1-mut or empty vector (MS2) (Fig. 4d). Likewise, lncRNA-TUBB2A, but not lncRNA-TUBB2A-mut RIP, was enriched for miR-3620-3p. Subsequent RNA pull-down experiments provided more solid evidence for these specific associations between the miRNAs and lncRNAs. LncRNA-TUBB2A, lncRNA-KRTAP5-AS1, lncRNA-TUBB2A-mut, and lncRNA-KRTAP5-AS1-mut were transcribed in vitro from pGEM-T-lncRNA-TUBB2A, pGEM-T-lncRNA-KRTAP5-AS1, pGEM-T-lncRNA-TUBB2A-mut and pGEM-T-lncRNA-KRTAP5-mut vectors, respectively. The in vitro transcription products were then labeled with biotin, incubated with cell lysates before isolation with streptavidin agarose beads and analyzed via real-time PCR. MiR-3620-3p could be pulled down by biotin-labeled lncRNA-TUBB2A and lncRNA-KRTAP5-AS1, but not lncRNA-TUBB2A-mut or

lncRNA-KRTAP5-AS1-mut. At the same time, biotin-labeled lncRNA-KRTAP5-AS1, but not lncRNA-KRTAP5-AS1-mut, could also bind miR-596 (Fig. 4e). These data reveal that lncRNA-KRTAP5-AS1 can bind miR-596 and miR-3620-3p. At the same time, lncRNA-TUBB2A can bind miR-3620-3p.

**LncRNAs act as ceRNAs to regulate cell function in vitro**. In order to test the biological functions of these two lncRNAs, we performed gain- and loss-of-function studies in GC cancer cells. We stably overexpressed or knocked down lncRNA-KRTAP5-AS1 in SGC-7901, AGS, and HGC-27 cells (Supplementary Fig. 8a, b). LncRNA-KRTAP5-AS1 overexpression led to increased proliferative capacity, as measured by the CCK-8 assay. When we explored its effect using transwell assays, lncRNA-KRTAP5-AS1 strongly promoted cell invasion ability. Not surprisingly, miR-596 and miR-3620-3p could abolish both biological functions caused by lncRNA-KRTAP5-AS1 (Fig. 5a–d, Supplementary Fig. 8c, d). In contrast, knockdown of endogenous lncRNA-KRTAP5-AS1 expression dramatically reduced the proliferative capacity and invasion ability of AGS cells. As expected, inhibition of miR-596 or miR-3620-3p could again decrease these effects (Fig. 5e–h, Supplementary Fig. 8e, f).

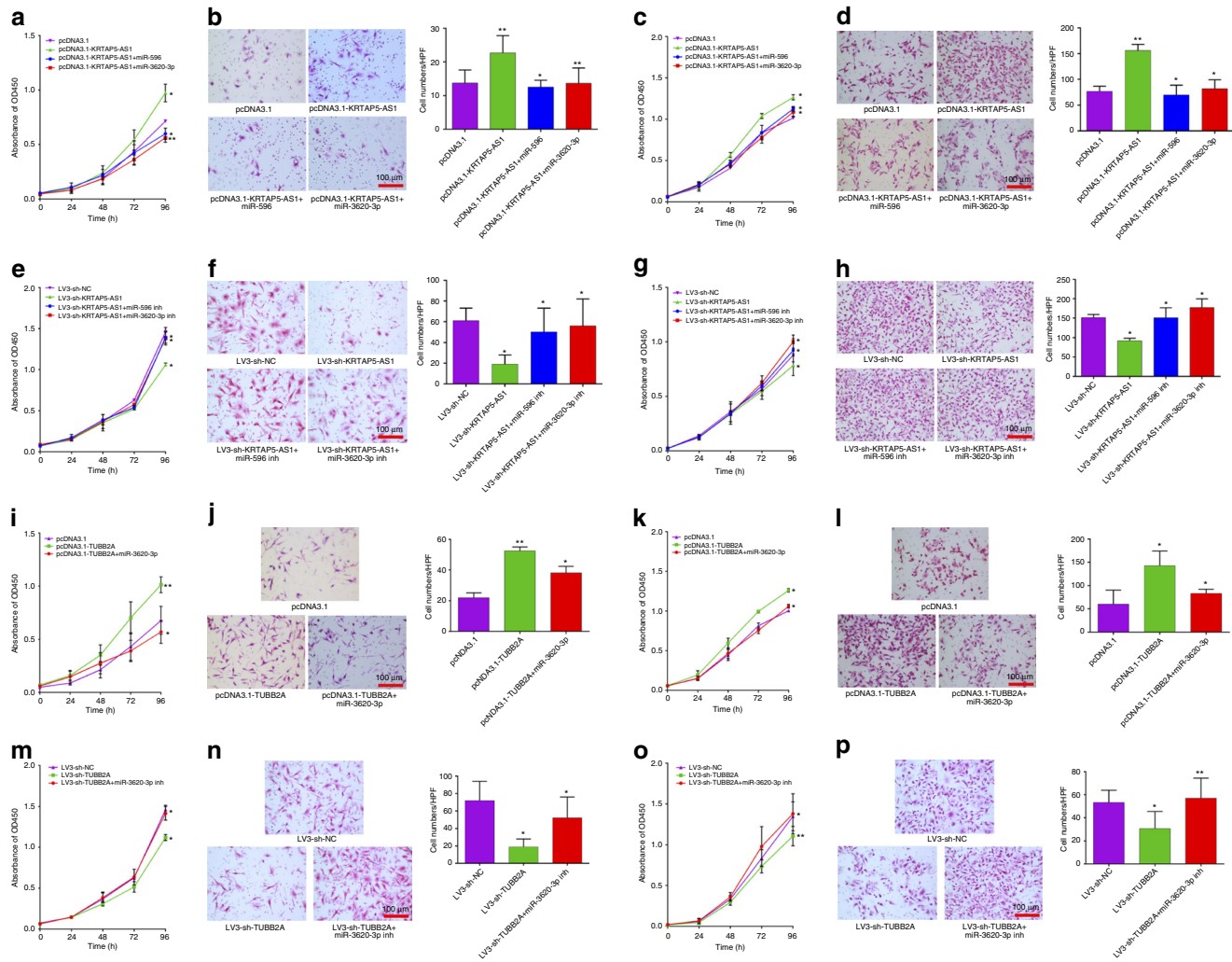

**Fig. 5** LncRNA-KRTAP5-AS1 and lncRNA-TUBB2A enhance the proliferative and invasive capacity of GC cells in vitro. **a** Cell proliferation was assessed daily for 4 days using the CCK-8 assay in lncRNA-KRTAP5-AS1 overexpressing SGC-7901 cells. **b** Transwell assays were used to evaluate the involvement of lncRNA-KRTAP5-AS1 for invasion in lncRNA-KRTAP5-AS1 overexpressing SGC-7901 cells. **c** Cell proliferation assessed in lncRNA-KRTAP5-AS1 overexpressing AGS cells. **d** Transwell assays assessed in lncRNA-KRTAP5-AS1 overexpressing AGS cells. **e** Cell proliferation assessed in lncRNA-KRTAP5-AS1 knockdown SGC-7901 cells. **f** Transwell assays assessed in lncRNA-KRTAP5-AS1 knockdown SGC-7901 cells. **g** Cell proliferation assessed in lncRNA-KRTAP5-AS1 knockdown AGS cells. **h** Transwell assays assessed in lncRNA-KRTAP5-AS1 knockdown AGS cells. **i** Cell proliferation assessed in lncRNA-TUBB2A overexpressing SGC-7901 cells. **j** Transwell assays assessed in lncRNA-TUBB2A overexpressing SGC-7901 cells. **k** Cell proliferation assessed in lncRNA-TUBB2A overexpressing AGS cells. **l** Transwell assays assessed in lncRNA-TUBB2A overexpressing AGS cells. **m** Cell proliferation assessed in lncRNA-TUBB2A knockdown SGC-7901 cells. **n** Transwell assays assessed in lncRNA-TUBB2A knockdown SGC-7901 cells. **o** Cell proliferation assessed in lncRNA-TUBB2A knockdown AGS cells. **p** Transwell assays assessed in lncRNA-TUBB2A knockdown AGS cells. In **b**, **d**, **f**, **h**, **j**, **l**, **n**, **p**, cells were incubated for 24 h, and counted under the microscope. Original magnification × 200. Scale bars = 100 μm. Data are shown as mean ± s.d., n = 3. The data statistical significance is assessed by Student's t-test. *P < 0.05, **P < 0.01

Similar experiments were performed to test the biological functions of lncRNA-TUBB2A by stable overexpression and suppression in SGC-7901, AGS, and HGC-27 cells (Supplementary Fig. 8g, h). Briefly, by CCK-8 and transwell experiments, lncRNA-TUBB2A overexpression was found to increase the proliferative capacity and invasion ability of GC cells, an effect which could be partially eliminated by miR-3620-3p (Fig. 5i–l, Supplementary Fig. 8i, j). In contrast, inhibition of miR-3620-3p could rescue the reduction in proliferative capacity and invasion ability caused by lncRNA-TUBB2A knockdown (Fig. 5m–p, Supplementary Fig. 8k, l).

To identify whether lncRNA-KRTAP5-AS1 and lncRNA-TUBB2A act as ceRNAs in the CLDN4 regulatory network, we transfected the luciferase reporter plasmid psiCHECK-CLDN4

into GC cells. LncRNA-KRATP5-AS1 could increase its luciferase activity, and this was abolished by miR-596 and miR-3620-3p. Similarly, miR-3620-3p could partially eliminate the effect of lncRNA-TUBB2A on increasing the luciferase activity of psiCHECK2-CLDN4 (Fig. 6a). Moreover, we performed RIP assays based on Ago2, which can enrich for targets bound by miRNAs upon immunoprecipitation. We separately overexpressed lncRNA-KRTAP5-AS1 and lncRNA-TUBB2A in SGC-7901 and AGS cells then pulled down Ago2 using an anti-Ago2 antibody. Overexpression of lncRNA-KRTAP5-AS1 or lncRNA-TUBB2A both caused a significant decrease in the enrichment of CLDN4 transcripts pulled down by Ago2 (Fig. 6b), indicating that there were less miRNA-bound CLDN4 transcripts present. This suggests that lncRNA-KRTAP5-AS1 and

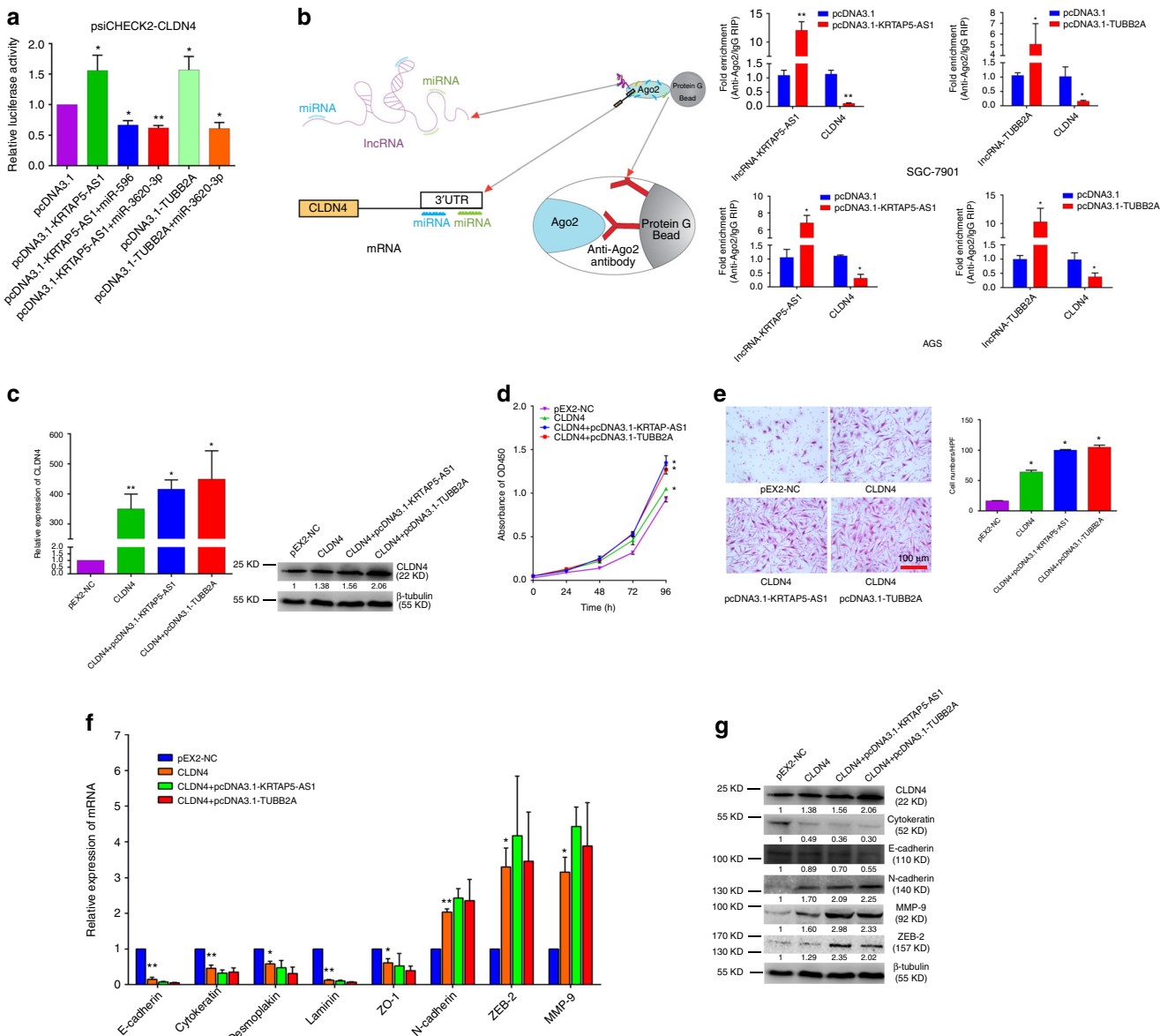

**Fig. 6** LncRNA-KRTAP5-AS1 and LncRNA-TUBB2A act as ceRNAs. **a** Luciferase activity in GC cells co-transfected with psiCHECK2-CLDN4 and ceRNAs. Data were presented as the relative ratio of renilla luciferase activity and firefly luciferase activity. **b** The *schematic diagram* and real-time PCR results of the RIP based on Ago2 showed that lncRNAs can compete with the CLDN4 transcript for the binding of miRNAs. **c** The relative expression levels of CLDN4 were determined by real-time PCR and western blotting after co-transfecting ceRNAs. **d** Cell proliferation was assessed daily for 4 days using the CCK-8 assay in CLDN4 overexpressed cells transfected the plasmids of pcDNA3.1-KRTAP5-AS1 and pcDNA3.1 -TUBB2A. **e** Transwell assays were used to assay the involvement of CLDN4 for invasion in CLDN4 overexpressing cells. Cells were incubated for 24 h, and counted under the microscope. Original magnification ×200. *Scale bars* = 100 μm. **f**, **g** The transcriptional and translational levels of EMT related markers. The real-time PCR and western blotting were performed at 48 h after the CLDN4 overexpressing cells treated with pcDNA3.1-KRTAP5-AS1 and pcDNA3.1-TUBB2A. Data are shown as mean ± s.d., n = 3. The data statistical significance is assessed by Student's t-test. *P < 0.05, **P < 0.01

lncRNA-TUBB2A can compete with the CLDN4 transcript for the binding of miRNAs. To shed more light on this aspect, we overexpressed these two lncRNAs into CLDN4 overexpressing cells. The expression of CLDN4, which could be down regulated by miR-596 and miR-3620-3p as we demonstrated above, could be enhanced at both the transcriptional and translational level by lncRNA-KRTAP5-AS1 or lncRNA-TUBB2A (Fig. 6c). The ability of CLDN4 to promote cell proliferation and invasion in vitro could be abolished by miR-596 and miR-3620-3p, as mentioned above, and overexpression of lncRNA-KRTAP5-AS1 or lncRNA-TUBB2A in CLDN4 cells significantly further increased these abilities (Fig. 6d, e). Subsequently, the EMT-inducing effects of

these two lncRNAs were measured. The downregulation of epithelial markers (E-cadherin, Cytokeratin) and the upregulation of mesenchymal markers and invasion related markers (N-cadherin, ZEB-2, MMP9) induced by CLDN4 were exaggerated after either lncRNA was overexpressed. These results were verified at both the transcriptional and translational levels (Fig. 6f, g). In summary, we found that through functioning as ceRNAs to regulate CLDN4 expression, lncRNA-KRTAP5-AS1 and lncRNA-TUBB2A can promote cell proliferation, invasion, and EMT. LncRNA-KRTAP5-AS1 appears to serve as a ceRNA for miR-596 and miR-3620-3p while lncRNA-TUBB2A serves as a ceRNA for miR-3620-3p.

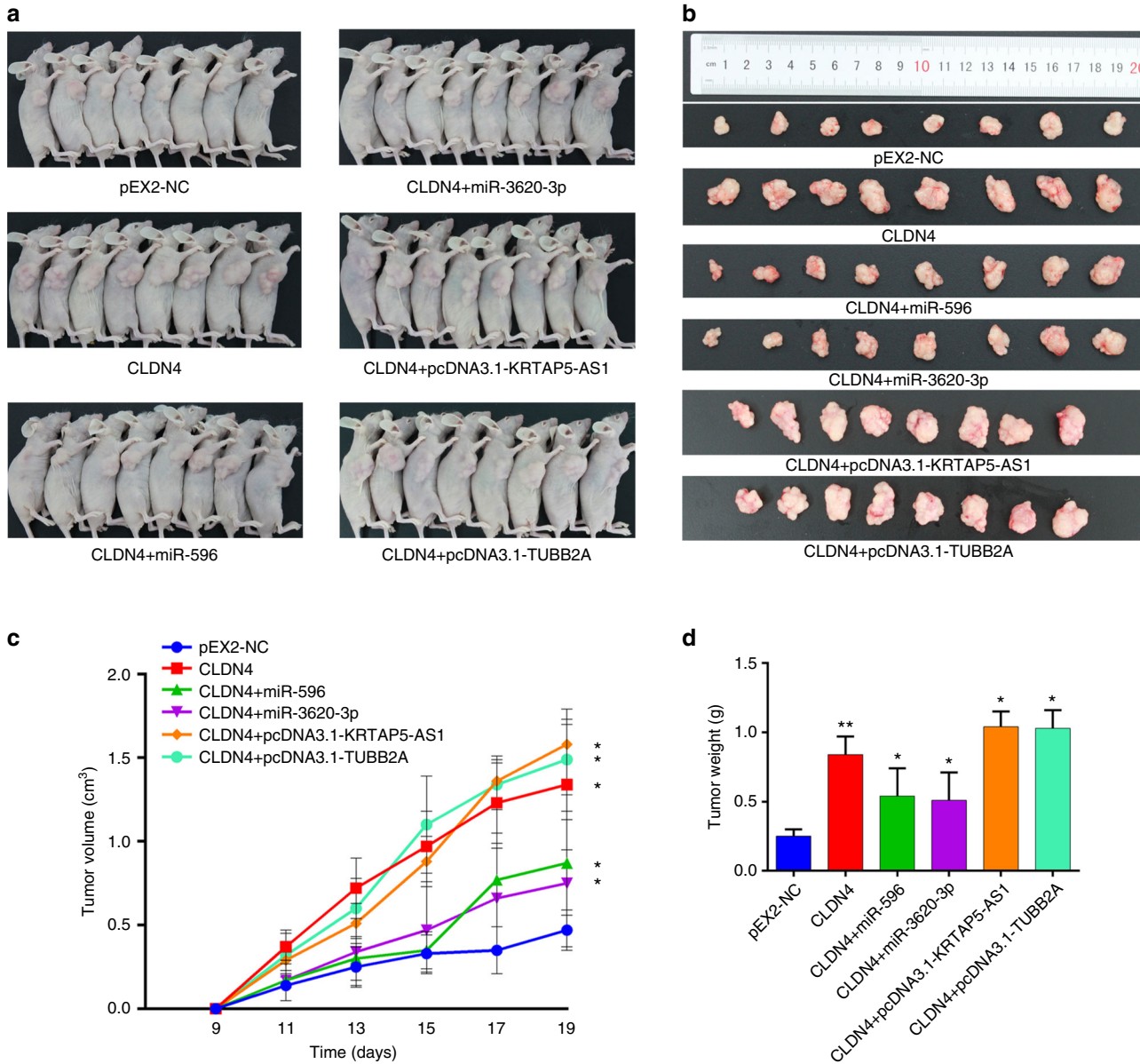

**Fig. 7** CLDN4 promotes proliferation in vivo. **a** In vivo tumor lumps of xenograft mouse models composed of CLDN4 overexpressing cells, which were treated with miR-596 or miR-3620-3p as well as pcDNA3.1-KRTAP5-AS1 or pcDNA3.1-TUBB2A. Mice were sacrificed at the 19th day after injection and each tumor lump was removed from the body. **b** *Images* of the tumor lumps of each group at the endpoint of the experiment described in **a**. **c** The tumor growth curves of in vivo tumor volumes. Data are mean ± s.d. of the tumor volumes, $n = 8$, *$P < 0.05$. **d** The mean tumor weight of each group. Data are shown as mean ± s.d. of the tumor weights, $n = 8$. The data statistical significance is assessed by Student's t-test. *$P < 0.05$, **$P < 0.01$

**Non-coding RNAs regulate the function of CLDN4 in vivo**. To evaluate the biological functions of these genes in vivo, different SGC-7901 or HGC-27 cells were subcutaneously or intravenously injected into nude mice. In total, there were six groups: Group 1 (pEX2-NC) was injected with pEX2-NC cells; Group 2 (CLDN4) was injected with CLDN4 overexpressing cells; Group 3 (miR-596 +CLDN4) was injected with CLDN4 overexpressing cells transfected with miR-596 mimics; Group 4 (miR-3620-3p+CLDN4) was injected with CLDN4 overexpressing cells transfected with miR-3620-3p mimics; Group 5 (lncRNA-KRTAP5-AS1+CLDN4) was injected with CLDN4 overexpressing cells transfected with lncRNA-KRTAP5-AS1; and Group 6 (lncRNA-TUBB2A +CLDN4) was injected with CLDN4 overexpressing cells transfected with lncRNA-TUBB2A. We found that tumor lumps in the CLDN4 group were significantly larger than in the pEX2-NC

group, and that miR-596 and miR-3620-3p could partially reduce the growth trend caused by CLDN4. Moreover, the tumor volumes in the lncRNA overexpressing groups were larger than in the CLDN4 overexpressing group. At the end of the experiment, the mice were sacrificed and we measured the volume and weight of the tumor lumps in each group (Fig. 7a, b). For SGC-7901 cells, the mean tumor volume at the time of death in mice injected with CLDN4 overexpressing cells was $1.34 \pm 0.39$ (mean value ± s.d.) cm$^3$ and the mean tumor volume of mice injected with NC cells was $0.47 \pm 0.12$ cm$^3$. The mean tumor volumes in the miR-596+CLDN4 and miR-3620-3p+CLDN4 groups were both smaller ($0.87 \pm 0.31$ cm$^3$ and $0.75 \pm 0.38$ cm$^3$, respectively) than in the CLDN4 overexpressing group. Moreover, the tumor volumes in the lncRNA-KRTAP5-AS1 +CLDN4 and lncRNA-TUBB2A+CLDN4 groups were larger

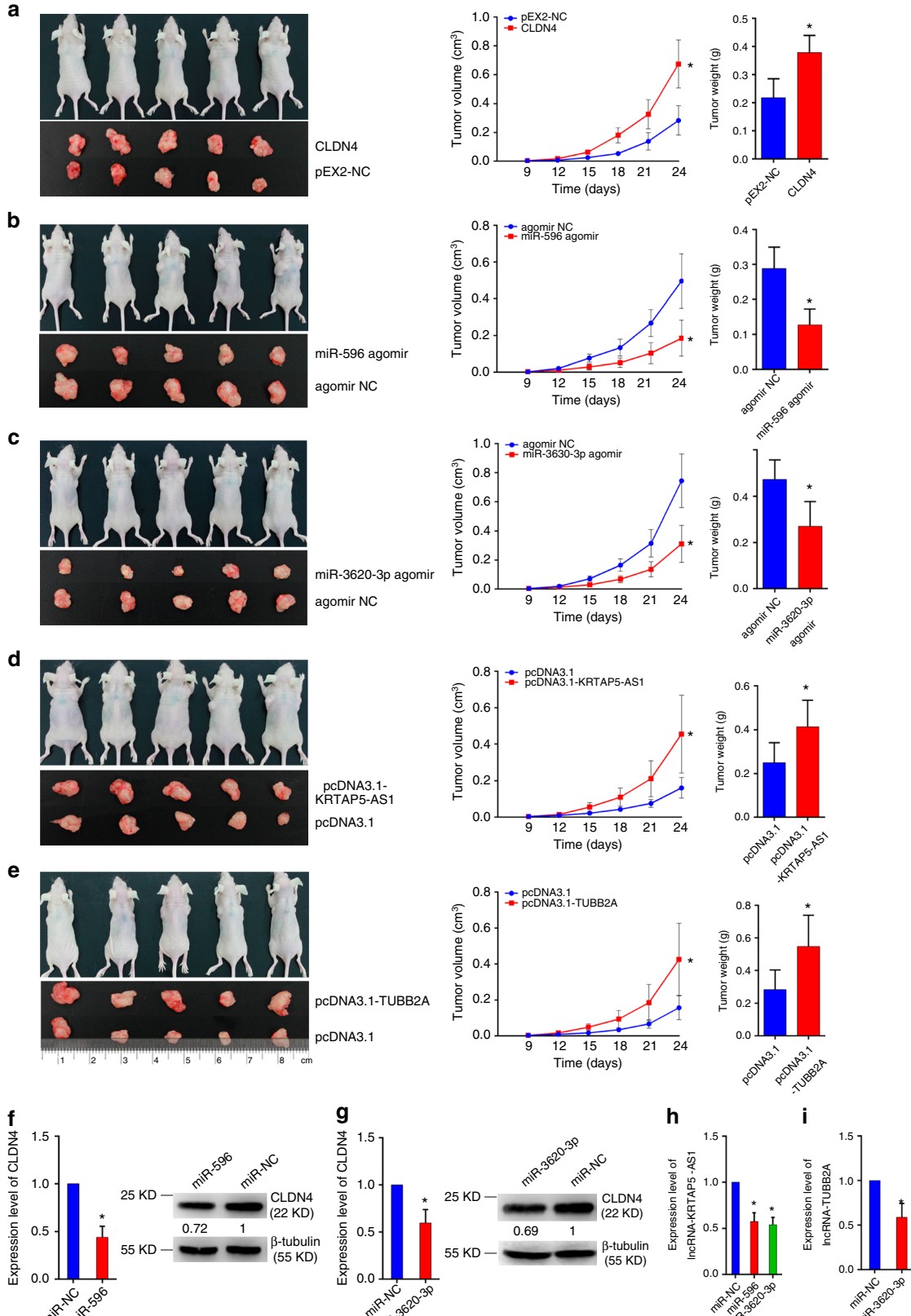

**Fig. 8** The effects of CLDN4 and ncRNAs on tumor growth. SGC-7901 cells were subcutaneously injected at both right and left armpit regions of nude mice. After tumor formation, plasmids of CLDN4, lncRNAs and agomirs of miRNAs were injected into tumors of right side and their negative controls into *left side*. **a** Treatment of CLDN4 promoted xenograft tumor growth. **b**, **c** Treatment of miR-596 and miR-3620-3p suppressed xenograft tumor growth. **d**, **e** Treatment of pcDNA3.1-KRTAP5-AS1 and pcDNA3.1-TUBB2A promoted xenograft tumor growth. **f**, **g** miR-596 and miR-3620-3p treated tumors showed less CLDN4 expression. **h** miR-596 and miR-3620-3p treated tumors showed less lncRNA-KRTAP5-AS1 expression. **i** miR-3620-3p treated tumors showed less lncRNA-TUBB2A expression. Data are shown as mean ± s.d., $n = 5$ for each group. The data statistical significance is assessed by Student's *t*-test. *$P < 0.05$, **$P < 0.01$

$(1.58 \pm 0.21 \, \text{cm}^3$ and $1.49 \pm 0.21 \, \text{cm}^3$, respectively) than in the CLDN4 group, which supported the findings of the in vitro experiments (Fig. 7c). Xenograft tumors grown from CLDN4 cells had greater mean weights than those grown from pEX2-NC cells $(0.84 \pm 0.13 \, \text{g}$ vs. $0.25 \pm 0.05 \, \text{g})$ (Fig. 7d). Also, the tumor weights were lower in the miR-596 group $(0.54 \pm 0.20 \, \text{g})$ and miR-3620-3p group $(0.51 \pm 0.20 \, \text{g})$ than in the CLDN4 over-expressing group, but much higher in the lncRNA-KRTAP5-AS1 +CLDN4 and lncRNA-TUBB2A+CLDN4 groups $(1.04 \pm 0.11 \, \text{g}$ and $1.03 \pm 0.13 \, \text{g}$, respectively). The experiments using HGC-27 cells showed similar effects as the SGC-7901 cells (Supplementary Fig. 9). Thus, consistent with our in vitro findings, we further found that the effect of CLDN4 on increasing tumor proliferation could be rescued by miR-596 or miR-3620-3p and enlarged when lncRNA-KRTAP5-AS1 or lncRNA-TUBB2A were overexpressed

in the nude mouse model. To test whether these effects were consistent once tumors had already formed, we subcutaneously injected the same amount of SGC-7901 cells into both sides of nude mice. Once the tumors had formed, we injected plasmids of CLDN4, the lncRNAs, or agomirs of the miRNAs into the tumors on one side and their negative controls into that on the other side. The injections were performed five times at an interval of 2 days between each injection (i.e., day 9, 12, 15…). 3 days after the last injection, the mice were sacrificed and the volumes and weights of their tumors were measured. The results showed that CLDN4, lncRNA-TUBB2A, and lncRNA-KRTAP5-AS1 could promote proliferation even after tumor formation compared with their negative controls and that tumor proliferation could be suppressed by miR-596 and miR-3620-3p (Fig. 8a–e). Furthermore, tumors injected with miR-596 agomir showed less CLDN4

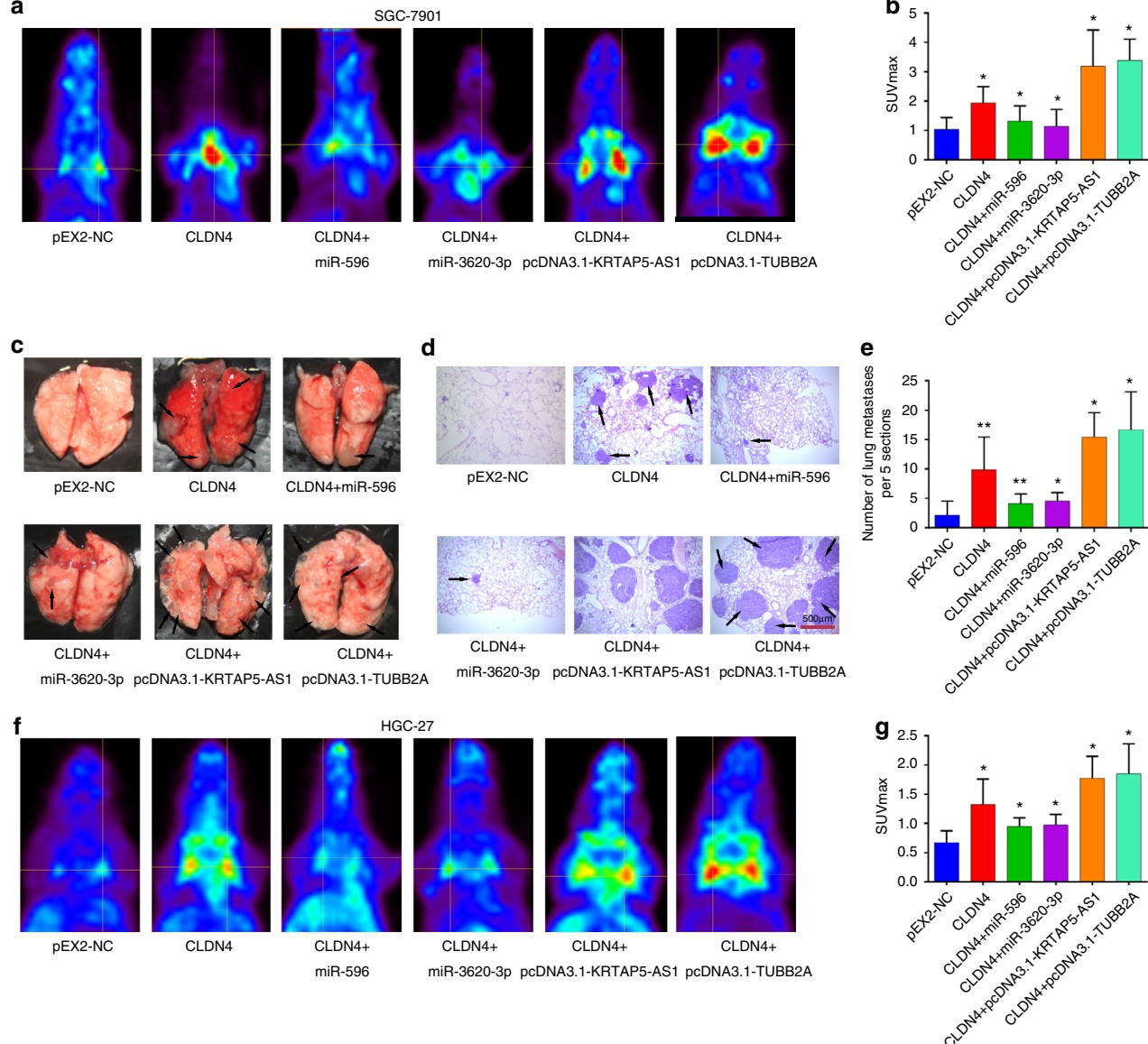

**Fig. 9** The metastasis promoting effect of CLDN4 could be regulated in vivo. **a, b** Transverse section of 18F-FDG PET images of mice at the 56 day after tail vein injection with $1 \times 10^6$ SGC-7901 cell clones and the max SUVs were analyzed in each group. Data are mean ± s.d. of the tumor volumes, $n = 8$, $*P < 0.05$. **c** The gross lesion of lung tissues isolated from the mice. **d** The microscopic images of lung tissue sections stained by hematoxylin and eosin. *Scale bars* = 500 μm. **e** The number of metastatic nodules in the lungs from 56 days after tail vein injection in **a** (five sections evaluated per lung). Data are mean ± s.d. of the tumor volumes, $n = 8$, $*P < 0.05$. **f, g** Transverse section of 18F-FDG PET images of mice at the 56 day after tail vein injection with $1 \times 10^6$ HGC-27 cell clones and the max SUVs were analyzed in each group. Data are shown as mean ± s.d. of the tumor volumes, $n = 8$. The data statistical significance is assessed by Student's *t*-test. $*P < 0.05$

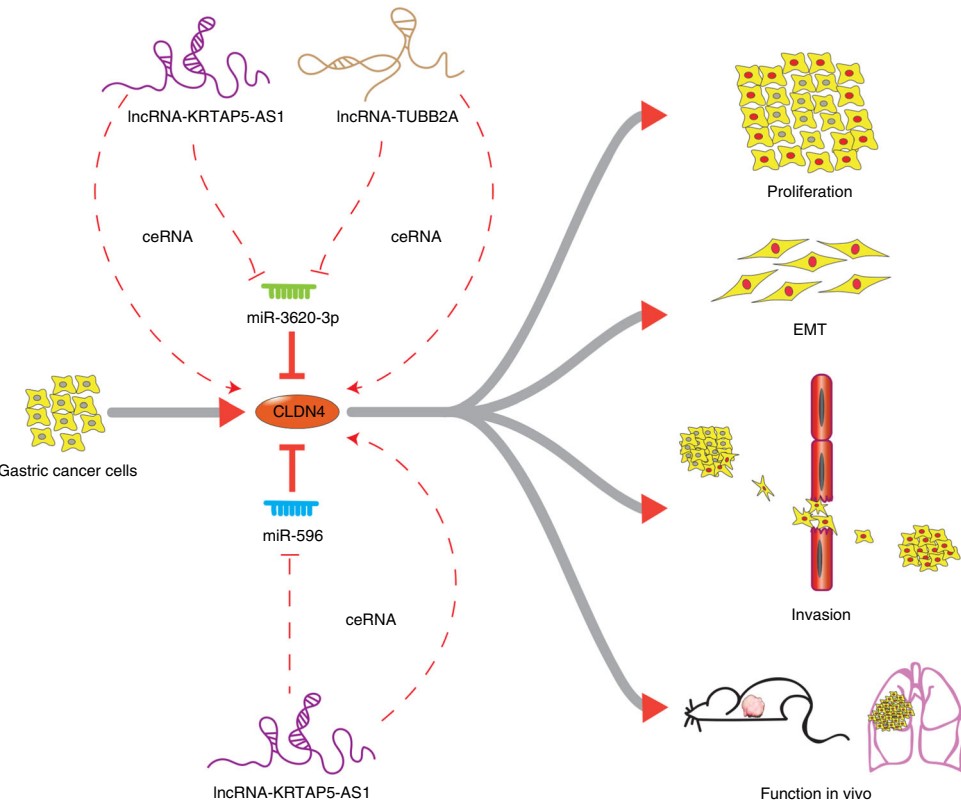

**Fig. 10** The mechanism graph of the regulatory network and function of CLDN4. CLDN4 could promote proliferation, metastasis or EMT processes of GC, which could be inhibited by miR-596, miR-3620-3p and enhanced by lncRNA-KRTAP5-AS1, lncRNA-TUBB2A as ceRNAs

expression on the transcriptional and translational levels and less lncRNA -KRTAP5-AS1 on the transcriptional level. Similarly, tumors injected with miR-3620-3p agomir showed less CLDN4, lncRNA-TUBB2A, and lncRNA -KRTAP5-AS1 expression (Fig. 8f–i).

To examine the influence these genes had on metastasis in vivo, the same six groups of cell clones were intravenously injected into nude mice via the tail vein. After 8 weeks of the injection, Positron Emission Tomography (PET) scanning was performed on each mouse. The results of the PET scans revealed that SGC-7901 cells overexpressing CLDN4 caused greater lung tumor formation, with a higher maximum standardized uptake value (SUVmax) of $1.94 \pm 0.55$ compared to pEX2-NC cells $(1.04 \pm 0.40)$. At the same time, the SUVmax in the miR-596 +CLDN4 and miR-3620-3p+CLDN4 groups were significantly smaller $(1.31 \pm 0.53$ and $1.14 \pm 0.58$, respectively) than those in the CLDN4 group. The SUVmax were $3.18 \pm 1.24$ in the lncRNA-KRTAP5-AS1+CLDN4 group and $3.38 \pm 0.73$ in the lncRNA-TUBB2A+CLDN4 group, both of which were larger than the CLDN4 group (Fig. 9a, b). Following PET scanning, mice were sacrificed and the morphological characteristics of their lungs were examined to support the PET scan results (Fig. 9c). We observed that the metastatic lesions at the surface of the lungs were more plentiful in the CLDN4 group than in the NC group, but more scarce and harder to observe in the miR-596 and miR-3620-3p+CLDN4 groups. Importantly, we also found that there were more metastatic lesions in the lncRNA+CLDN4 groups compared with the CLDN4 group. Afterwards, hematoxylin-eosin staining was performed on each lung and the numbers of metastatic nodules were counted under the microscope. Consistent with the PET scan results and morphological characteristics, there were significantly more metastatic nodules in the lungs of mice injected with CLDN4 overexpressing cells compared with pEX2-NC cells $(9.89 \pm 5.53$ vs. $2.14 \pm 0.56)$. The average number of metastatic nodules was $4.11 \pm 1.64$ in the miR-596+CLDN4 group and $4.57 \pm 1.38$ in the miR-3620-3p +CLDN4 group, showing a rescue of the tumor-promoting effects caused by CLDN4 overexpression. For lncRNA-KRTAP5-AS1 +CLDN4 and lncRNA-TUBB2A+CLDN4 groups, the numbers of metastatic nodules were $15.43 \pm 4.15$ and $16.66 \pm 6.45$, respectively, with both increased compared to the CLDN4 group (Fig. 9d, e). Verification experiments performed in HGC-27 cells further supported the above observations (Fig. 9f, g). In summary, we used subcutaneously transplanted tumor models and tumor metastasis models to test the effects of different cell clones on proliferation and metastasis in vivo. Through PET scanning, morphological observation, and hematoxylin-eosin staining, we revealed that CLDN4 promotes proliferation and metastasis in vivo, which could be inhibited by miR-596 and miR-3620-3p and enhanced by lncRNA-KRTAP5-AS1 and lncRNA-TUBB2A.

**Discussion**

With the development of molecular biology techniques, many molecular mechanisms of GC are now being revealed[43, 44]. Thousands of genes have already been demonstrated to play important roles in many cancer processes. It is known that genes generally do not function alone, so they can be grouped into "networks" based on their interactions. In this study, we used microarrays and RNA-sequencing to identify the regulatory networks of mRNAs and ncRNAs in GC. Among the results of the analysis for potential mRNA–ceRNA interactions in the "Tight junction" pathway, CLDN4 goaded our interest since it is closely related to cancer development.

Although there is still some controversy regarding the role of CLDN4 in carcinogenesis[36], many studies have revealed

oncogenic functions for CLDN4 such as enhancing proliferation, invasion, and EMT[37, 38]. Moreover, we have previously demonstrated aberrant expression of CLDN4 in GC and precursor lesions[35]. In our current work, we validated the oncogenic function of CLDN4 in reinforcing the proliferative, invasive, and metastatic capacities of GC cells and promoting EMT through a series of functional experiments both in vitro and in vivo. The oncogenic role of CLDN4 in enhancing invasion and EMT is supported by previous studies showing that claudin family members can activate MMP and ZEB family members[38, 45]. In addition, the proliferation and metastasis promoting effects of CLDN4 in vivo can be partially explained by its ability to promote vasculogenic mimicry formation[46]. Up to now, studies have shown that epigenetic alterations may directly lead to the aberrant expression of CLDN4[47, 48], but the regulatory mechanisms of CLDN4 still remained to be elucidated in GC, especially regarding the role of ncRNAs in CLDN4 regulation. Given the significant effects of CLDN4 on cancer metastasis, as revealed by our work and others, the regulatory network related to CLDN4 urgently needed to be explored. In this study, we found that CLDN4, a common target of miR-596 and miR-3620-3p, can significantly enhance the proliferation and invasion of GC cells both in vitro and in vivo. These effects of CLDN4 are decreased upon exogenous introduction of miR-596 and miR-3620-3p. We also observed that CLDN4, whose expression is negatively associated with the expression of miR-596 and miR-3620-3p, is up-regulated in GC tissues and significantly correlates with poor survival for GC patients. In contrast with the inhibitory role of the above miRNAs on CLDN4, our results show that exogenous introduction of lncRNA-KRTAP5-AS1 and lncRNA-TUBB2A can elevate the effects of CLDN4 on proliferation and invasion. All of these results drove us to propose the existence of a noteworthy regulatory network in which miRNAs and lncRNAs interact with each other to co-regulate the expression pattern and function of CLDN4 (Fig. 10).

In recent years, the functions of ncRNAs have drawn more and more attention[49]. The functions and regulatory mechanisms of miRNAs are continuously becoming more deeply understood. Nevertheless, we present that the expression and function of CLDN4 can be negatively regulated by two miRNAs, miR-596, and miR-3620-3p, both of which were largely enigmatic before this study. Unlike miRNAs, the functions of lncRNAs remain largely unknown. Accumulating evidence has indicated that lncRNAs can function as ceRNAs for miRNAs in cancer[50]. For example, Yuan et al. showed that lncRNA-ATB functions as a ceRNA for miR-200s, thereby regulating the expression of ZEB1 and ZEB2 in hepatic cellular carcinoma[51]. Liu et al[52]. showed that lncRNA-HOTAIR can regulate HER2 expression by acting as a sponge for miR-331-3p in GC. Likewise, LncRNA-ARSR can act as a ceRNA for miR-34 and miR-449 to facilitate AXL and c-MET expression, thus promoting sunitinib resistance in renal cell carcinoma[53]. Here, we show that lncRNA-KRTAP5-AS1 acts as an oncogene in vitro through binding miR-596 and miR-3620-3p and lncRNA-TUBB2A functions similarly through binding miR-3620-3p. This indicates that these two lncRNAs can serve as ceRNAs. Subsequent investigations such as luciferase activity assays, RIP based on Ago2 and in vivo experiments further confirmed that these two lncRNAs function as ceRNAs to regulate CLDN4.

As shown above, ncRNAs may regulate multiple targets in different cells using different binding regions. Moreover, one gene can be regulated by multiple ncRNAs. Thus, ncRNAs and mRNAs can build complicated networks. Although we demonstrate that the effects of CLDN4 on invasion and proliferation are fully due to its direct interaction with miR-596 and miR-3620-3p, our results indicate that these two miRNAs may also be involved

in other regulatory circuits. During examination of cell apoptosis, we found that miR-596 and miR-3620-3p could significantly promote GC cell apoptosis, whereas there was no obvious difference in apoptosis between CLDN4 overexpressing cells and pEX2-NC cells. This regulatory function of miR-596 on cell apoptosis is in accordance with the results of a previous study[54]. As it is well known that a single miRNA can bind multiple target mRNAs through miRNA response elements (MREs)[23, 55, 56], it is rational that miR-596 and miR-3620-3p could regulate cell apoptosis through targeting additional transcripts in addition to the CLDN4 regulatory network. Likewise, although the lncRNA-KRTAP5-AS1 and lncRNA-TUBB2A can interact with miR-596 and miR-3620-3p to regulate EMT, we observed some differences among the EMT markers. These multiplex results are supported by studies showing that lncRNAs, which possess the capacity to function as ceRNAs, can target MREs for various mRNA-targeting miRNAs, similar to how miRNAs can target multiple mRNAs[55, 56]. These findings imply that the lncRNAs may also participate in some additional regulatory networks. Therefore, further studies are warranted to deeply investigate these additional regulatory networks.

In summary, we reported a regulatory network for CLDN4 in GC. We observed that CLDN4 was up-regulated in GC and associated with poor prognosis. CLDN4 reinforced proliferation, invasion, and EMT in GC cells, which could be reversed by miR-596 and miR-3620-3p. In addition, lncRNA-KRTAP5-AS1 and lncRNA-TUBB2A could act as ceRNA to increase these functions of CLDN4. These results reveal that ncRNAs play important roles in the regulatory network of CLDN4. As such, ncRNAs should be considered as potential biomarkers and therapeutic targets against GC.

## Methods

**Tissues**. One hundred and four GC tissues and matched non-tumorous adjacent tissues were obtained from patients who were newly diagnosed and received surgical resection at the First Affiliated Hospital of China Medical University, between 2007 and 2011 (detailed information seen in Supplementary Data 10). Informed consent was obtained from all patients enrolled in this study. Histological grade was staged according to the seventh TNM staging of the International Union against Cancer (UICC)/American Joint Committee on Cancer (AJCC) system. All research complied with the principles of the Declaration of Helsinki, and approval was acquired from the Research Ethics Committee of the First Affiliated Hospital of China Medical University.

**Cell culture**. SGC-7901, HGC-27 (purchased from the Institute of Biochemistry and Cell Biology at the Chinese Academy of Sciences, Shanghai, China), and AGS cells (obtained from ATCC, Manassas, VA) were cultured at 37 °C in RPMI 1640 medium (Invitrogen, Carlsbad, CA, USA) supplemented with 10% fetal bovine serum in a humidified incubator in an atmosphere containing 5% $CO_2$ (Thermo, Waltham, MA, USA). None of cell lines used in this paper were listed in the database of commonly misidentified cell lines maintained by ICLAC. All cell lines were free of mycoplasma contamination.

**Microarray analysis**. Six match-paired sets of tissues for microarray were obtained from patients who were newly diagnosed with GC and received radical resection at the First Affiliated Hospital of China Medical University. Total RNA was extracted from the above tissues. The Quick Amp Labeling kit (Agilent Technologies, Palo Alto CA, USA) was used to amplify and transcribe the RNA into cRNA, then the labeled cRNA was hybridized onto the Human LncRNA Array v3.0 (8 × 60 K, ArrayStar, Rockville, MD, USA) using the Agilent Gene Expression Hybridization Kit (Agilent Technologies). Total miRNAs were labeled using the miRCURY™ (Hy3TM/Hy5TM) power labeling kit (Exiqon Life Sciences, Vedbaek, Denmark) and hybridized onto the miRCURY LNA microRNA array (v18.0) (Exiqon Life Sciences). The arrays were scanned with an Axon GenePix 4000B microarray scanner following the washing step. The acquired array images were extracted and analyzed using Agilent Feature Extraction Software v10.7. Raw signal intensities were normalized in a quantile method by GeneSpring GX v11.5.1 (Agilent Technologies), and low intensity lncRNAs, mRNAs, and miRNAs were filtered. LncRNAs, mRNAs, and miRNAs that were significantly differentially expressed were identified using box plot and scatter plot filtering. The threshold used to screen upregulated or downregulated lncRNAs, mRNAs and miRNAs was fold change > 2.0 with a $P$-value < 0.05.

**RNA isolation and real-time PCR**. Total RNA was extracted using Trizol reagent (Invitrogen). Poly-A tails were added to the miRNA according to the protocol of the Poly (A) Tailing Kit (Ambion, Waltham, MA, USA). The PrimeScriptTM RT reagent Kit with gDNA Eraser (Takara, Dalian, China), and gene-specific primers or random primers were used to generate cDNA. Real-time PCR was performed in a Light Cycler 480 II Real-Time PCR system (Roche Diagnostics, Basel, Switzerland) using SYBR® Green (Takara). Glyceraldehyde phosphate dehydrogenase (GAPDH) and U6 snRNA were employed as endogenous controls for mRNA/lncRNA and miRNA, respectively. The comparative Ct method was used to calculate the relative expression of RNAs. Primer sequences are displayed in Supplementary Data 11.

**Luciferase reporter assay**. Luciferase reporters were generated based on the psiCHECK2 vector (Promega). To construct psiCHECK2-CLDN4, the complete 3′ UTR of human CLDN4 mRNA (853nt, UCSC accession no. uc003tzh.2), containing the putative miR-596 and miR-3620-3p binding sites, was amplified and cloned into the psiCHECK2 vector. For the lncRNAs, full-length sequences of lncRNA-TUBB2A (1052nt, UCSC accession no. uc011dhu.1) and lncRNA-KRTAP5-AS1 (2554nt, UCSC accession no. uc001ltt.1) were PCR amplified and cloned into the psiCHECK2 vector. The luciferase reporter was co-transfected with miR-596 mimics, miR-3620-3p mimics, miR-4292 mimics, miR-596-mut mimics, miR-3620-3p-mut mimics, or miR-NC into SGC-7901 cells by Lipofectamine 2000 according to the manufacturer's guidelines. The relative luciferase activity was measured with the Dual-Luciferase Reporter Assay System (Promega) and Infinate M200 PRO microplate reader (Tecan, Shanghai, China).

**Western blotting analysis**. Total cell lysates were obtained using the Total Protein Extraction Kit (KeyGen Biotech). Proteins were separated by SDS-polyacrylamide gel electrophoresis (SDS–PAGE) and transferred to PVDF membranes (Millipore). Membranes were immunoblotted with primary antibodies (Supplementary Table 3). After incubation with peroxidase-conjugated affinipure goat anti-mouse IgG or peroxidase-conjugated affinipure goat anti-rabbit IgG, the blots were detected using the GelCapture version software (DNR Bio-Imaging Systems, Jerusalem, Israel).The loading control for the western blotting was β-tubulin. Main uncropped immunoblots are provided in Supplementary Fig. 10.

**Transfection**. Transfections were performed using the Lipofectamine 2000 Reagent (Invitrogen) following the manufacturer's protocol. Final concentrations of 50 nM of miRNA mimics and 0.75 μg/ml plasmids were used for each transfection in a six well plate with 2 ml culture medium. For lentivirus transfection, cells were transfected with $5 \times 10^6$ transducing units of lentivirus. The stable cell lines were constructed using G418 (200 μg ml$^{-1}$) or puromycin (200 μg ml$^{-1}$). Sequences for shRNAs and RNA oligoribonucleotides are listed in Supplementary Tables 4, 5.

**Cell Counting Kit-8 proliferation assay**. The capacity for cellular proliferation was measured using the Cell Counting Kit-8 (CCK-8) (Dojindo Laboratories, Kumamoto, Japan) according to the manufacturer's instructions. The optical density was determined with a microplate reader (Bio-Rad, Hercules, CA, USA) at a wavelength of 450 nm.

**Transwell invasion assay**. The transwell invasion assay was conducted using the transwell (Corning, NY, USA) and matrigel (BD Biosciences, San Jose, CA, USA) according to the manufacturer's instructions. Cells ($\sim 5 \times 10^4$) were added into the upper compartment of the chamber. After 24 h of incubation at 37°C with 5% $CO_2$, the number of cells invading through the matrigel was counted in 10 randomly selected visual fields from the central and peripheral portions of the filter using a Leica DM3000 microscope (Leica, Wetzlar, Germany).

**Scrape motility assay**. Scrape motility assays were performed by scratching the cell monolayer with a sterile 200 μl pipette tip, and the scratched areas were photographed at ×100 magnification using Leica DMI3000B computer-assisted microscope (Leica). Images were captured at 0, 6, 12, and 24 h after the scratch was made. Images were analyzed using Image-Pro Plus v6.0 image analysis software (Media Cybernetics, Rockville, MD, USA).

**Fluorescence-activated cell sorting**. Cells were trypsinized and resuspended to generate single-cell suspensions. For cell-cycle analysis, detached cells were fixed overnight in 70% ethanol at 4 °C, stained with propidium iodide in a cell cycle detection kit (KeyGen Biotech, Nanjing, China) according to the manufacturer's instructions, and analyzed using a FACS calibur flow cytometer (BD Biosciences) and BD Cell Quest software. To study apoptosis, cells were stained with Annexin V-APC and propidium iodide (Annexin V-APC Apotosis Detection Kit, KeyGEN) according to the manufacturer's instructions. Samples were analyzed using the LSRFortessa (BD Bioscience).

**Immunofluorescence analysis**. Different cells were cultured and fixed in a 48-well culture board and incubated with antibodies specific for E-cadherin (1:200, Abcam, ab76055), Cytokeratin (1:250, Abcam, ab53280), N-cadherin (1:200, Abcam, ab76011), or MMP-9(1:200, Abcam, ab119906). After incubating with goat anti-rabbit IgG (1:1000, Alexa Fluor594, Invitrogen, A-21207) and goat anti-mouse IgG (1:1000, Alexa Fluor594, Invitrogen, A-11032), the nuclei were stained by adding DAPI (1:1000, Invitrogen, D3571). Finally the cells were observed via Leica DMI3000 B (Leica).

**RNA-sequencing**. Six groups of SGC-7901 cells (purchased from the Institute of Biochemistry and Cell Biology at the Chinese Academy of Sciences, Shanghai, China) were cultured and then separately transfected with miR-596 mimics, miR-3620-3p mimics, miR-NC, pcDNA3.1-KRTAP5-AS1, pcDNA3.1-TUBB2A, or pcDNA3.1 before being prepared for massive RNA sequencing. Total RNA from the cell lines was extracted using Trizol reagent (Invitrogen), and gene-specific primers or random primers were used to generate cDNA. After quantitative analysis and quality inspection, we constructed sequencing libraries using the KAPA Stranded RNA-Seq Library Prep Kit (Illumina). RNA-sequencing was performed on an Illumina HiSeq 4000 Sequencing System with 150 cycles. After the data preprocessing, gene level fragments per kilobase of exon per million fragments mapped (FPKM)[57] were calculated. Details regarding the process of the RNA-sequencing and data analysis were listed in the Supplementary Methods.

**RNA pull-down**. LncRNA-TUBB2A and lncRNA-KRTAP5-AS1 were transcribed in vitro from pGEM-T-lncRNA-TUBB2A or pGEM-T-lncRNA-KRTAP5-AS1 vectors, respectively. Simultaneously, LncRNA-TUBB2A-mut and lncRNA-KRTAP5-AS1-mut were transcribed from pGEM-T-lncRNA-TUBB2A-mut or pGEM-T-lncRNA-KRTAP5 -AS1-mut vectors with their corresponding miRNA binding sites mutated. Altogether with transcripts transcribed from pGEM-T (used as the negative control), all five transcripts were biotin-labeled with the Biotin RNA Labeling Mix (Roche, Basel, Switzerland) and T7 RNA polymerase (Roche), treated with RNase-free DNase I (Roche), and purified with an RNeasy Mini Kit (Qiagen, Valencia, CA, USA). One milligram of whole-cell lysates from SGC-7901 cells or AGS cells was incubated with three micrograms of purified biotinylated transcripts for 1 h at 25 °C. The complexes were isolated by streptavidin agarose beads (Invitrogen). Any RNA present in the pull-down material was detected by real-time PCR analysis.

**RNA immunoprecipitation**. SGC-7901 and AGS cells were co-transfected with pcDNA3.1-MS2, pcDNA3.1-MS2- TUBB2A, pcDNA3.1-MS2-KRTAP5-AS1, pcDNA3.1-MS2-TUBB2A-mut, pcDNA 3.1-MS2-KRTAP5-AS1-mut, or pMS2-GFP (Addgene). After 48 h, the transfected cells were used in RNA immunoprecipitation (RIP) experiments using the Magna RIP$^{TM}$ RNA-Binding Protein Immunoprecipitation Kit (Millipore, Bedford, MA, USA) and an anti-GFP antibody (Roche) according to the manufacturer's instructions. The RNA fraction isolated by RIP was quantified using a Nano-Photometer spectrophotometer in the UV and visible spectra (Implen, Munich, Germany). Real-time PCR was used to evaluate the expression levels of miRNAs.

**RIP based on Ago2**. SGC-7901 and AGS cells were transfected with pcDNA3.1-KRTAP5-AS1, pcDNA3.1-TUBB2A, or pcDNA3.1. After 48 h, cells were used to perform RIP experiments using an anti-Ago2 antibody (Millipore, Bedford, MA, USA) and the Magna RIP$^{TM}$ RNA-Binding Protein Immunoprecipitation Kit (Millipore, Bedford, MA, USA), according to the manufacturer's instructions. RNAs were isolated from the immunoprecipitation products and quantified by a Nano-Photometer spectrophotometer in the UV and visible spectra (Implen, Munich, Germany). Real-time PCR was performed to examine the expression levels of lncRNAs and CLDN4.

**In situ hybridization**. In situ hybridization (ISH) was performed by applying the ISH Kit (Boster, Bio-Engineering Company, Wuhan, China). Formalin-fixed paraffin embedded (FFPE) tissue slides were deparaffinized and deproteinated. Slides were then prehybridized in prehybridization solution for 2 h at 42 °C and incubated with DIG-labeled probe solution over night at 42 °C. After stringent washing, the slides were then exposed to a streptavidin-peroxidase reaction system and stained with 3, 3′diaminobenzidine (DAB, ZSGB-BIO, Beijing, China). Hematoxylin was used to counterstain the slides.

**Animal experiments**. $2 \times 10^6$ SGC-7901 or HGC-27 cells in 0.2 ml PBS were subcutaneously injected into the right armpit region of 48 five-week-old female BALB/c nude mice which were randomly divided into six groups ($n = 8$ for each group). The tumor size was measured every 2 days with calipers. Nineteen days after injection, the mice were sacrificed and the subcutaneous tumors were isolated and measured.

For experiments in existing tumors, $2 \times 10^6$ SGC-7901 cells in 0.2 ml PBS were subcutaneously injected at both right and left armpits regions of 25 nude mice which were randomly divided into five groups ($n = 5$ for each group). After tumors formed, 10 μg plasmid of CLDN4 and lncRNAs or 1.5 nmol miRNA agomir of

miRNAs were injected into the tumors on the right side with their negative controls injected into the left side. The injections were performed five times at an interval of 2 days between each injection (i.e., day 9, 12, 15…). Tumor formation in each mouse was monitored every 3 days before the injection of plasmid or agomir by taking two-dimensional measurements of the tumor. After 3 days of the last injection, the mice were sacrificed and the volume and weight of their tumors were measured. The tumor volume was calculated with the formula: $(L \times W^2)/2$, where L is the length and W is the width of the tumor.

For metastasis experiments, $1 \times 10^6$ SGC-7901 or HGC-27 cells in 0.1 ml PBS were injected into the tail vein of 48 five-week-old female BALB/c nude mice which were randomly divided into six groups ($n = 8$ for each group). After 8 weeks of injection, 18-fluorodeoxyglucose Positron Emission Tomography (18F-FDG PET) scans were performed using a PET scanner (Metis 1800, Madic Technology Co, Ltd). Before 18F-FDG administration, the mice were fasted for at least four hours. Each mouse was injected intravenously with 7–12 MBq of 18F-FDG via the tail vein. The 18F-FDG was metabolized for 30 min after injection and then PET scanning was performed for 30 min. Afterwards, the PET images were carefully evaluated by two experienced nuclear medicine physicians. For semi-quantitative analysis, the maximum standardized uptake value (SUVmax) was measured and calculated according to the following formula:

$$SUV = \text{The radioactive concentration in the tumor lesion (MBq/g)} \times \text{Body weight of mice (g)/The injected dose of 18F-FDG (MBq)}.$$

After the 18F-FDG PET scan, intact lung tissues were isolated from the mice and tissue sections were stained with hematoxylin and eosin. The numbers of metastatic cancer nests were counted at $10 \times 10$ magnifications using an inverted microscope (Leica DMI300B).

For all animal experiments, the operators and investigators were blinded to the group allocation. All experimental procedures involving animals were done in accordance with the Guide for the Care and Use of Laboratory Animals (NIH publication no. 80-23, revised 1996) and the institutional ethical guidelines for animal experiments.

**Statistical analysis**. All statistical analyses in this study were performed using SPSS 20.0 software (IBM Corp, Armonk, NY, USA). Data are listed as mean value ± s.d. Student's t-test was used when the variance between groups are similar, and the Wilcoxon signed rank test was used when the variance between groups are not similar The Kaplan–Meier method with log-rank test, Spearman's correlation analysis, and Cox multivariate analysis were used as mentioned above. The tumor marker prognostic analysis was performed following the REMARK reporting guidelines[58]. A P-value <0.05 from a two-tailed test was considered significant.

**Data availability**. The microarray data are deposited in the NCBI Gene Expression Omnibus (GEO) datasets under the accession number GSE99417. The RNA sequence data generated in this study are deposited in the NCBI Sequence Read Archive (SRA) database under the SRA number SRP106526. The authors declare that all the data supporting the findings in this study are available in this study and its Supplementary Information, or are available from the corresponding author through reasonable request.

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

## Acknowledgements

We thank the department of Surgical Oncology of the First Affiliated Hospital of China Medical University for providing human gastric tissue samples. We also acknowledge Prof. Jason C. Mills and Joseph Burclaff from Washington University School of Medicine for the help with the English in the manuscript. This work was supported by the National Science Foundation of China (81372549), Ministry of Education team Development plan (IRT13101), the Special Prophase Program for National Key Basic Research Program of China (No. 2014CB560712) and Clinical Capability Construction Project for Liaoning Provincial Hospitals (LNCCC-A01-2014).

## Author contributions

Y.-X.S. and Z.-N.W. conceived the project and designed the experiments. Y.-X.S., J.-X.S. and J.-H.Z. started and performed majority of the experiments, with contributions from Y.-C.Y., J.-X.S., Z.-H.W. and Z.-F.M. Y.-X.S., J.-X.S. and J.-H.Z. analyzed results and wrote the manuscript. P.G. and X.-W.C. performed bioinformatics analyses and contributed to data analysis. Y.-X.S. and Z.-N.W. contributed to manuscript revision and supervised all experiments. All authors read and provided suggestions during manuscript preparation.
