## [Peer Review File · Nature Communications]

Reviewers' comments:

Reviewer #1 (Remarks to the Author):

In the current manuscript, using microarray-based analysis of mRNA, miRNA and lncRNAs from gastric cancer cells, the authors have identified plausible regulatory gene networks, comprising miRNA, mRNA and lncRNAs, that control gastric cancer development and progression. Among the enriched regulatory networks, authors have chosen tight junction pathway for further investigations. Tight junction pathway contains one of the critical genes Claudin 4 (CLDN4), which has been widely implicated in cancer metastasis and progression in several cancers. Here, the authors show that miR-596 and miR-3620-3p attenuate the effects of CLDN4 and interestingly lncRNAs such as lncRNA-KRTAP5-AS1 and lncRNA-TUBB2A promote the effects of CLDN4 by acting as ceRNAs for miR-596 and miR-3620-3p.

Although the current work is interesting, I have several concerns which I am listing below.

1. The major concern of the current work is the specificity of the regulatory interactions. For example, miR-596 and miR-3620-3p can induce apoptosis and cell cycle perturbations independent of the CLDN4 activity. lncRNAs also perform functions independent of CLDN4. This indicates that partial rescue of CLDN4 mediated functions could be due to their influence on the pathways that are independently controlled by miRNAs and lncRNAs. Likewise, can the authors claim that the effects on invasion and proliferation are fully due to the direct interaction of CLDN4 with these two miRNAs, and not as result of combined effect on other targets with the same function? For example, these 2 miRNAs are also predicted to target CLDN7, a gene product that has been identified by the authors as upregulated in gastric cancer and also part of the tight junction network.

2. The authors should try to verify a synergic effect of the 2 miRNAs on CLDN4 expression and function if transfected simultaneously?

3. It would be better if authors provide additional data to reinforce the specificity of these regulatory interactions. By performing RNA-seq, authors can characterize molecular pathways that are controlled by miRNA, lncRNA and CLDN4 in gastric cancer cell lines. This unbiased strategy will enable us to know whether these RNAs regulate common pathways. Authors should also provide in vivo evidence on mRNA-miRNA and lncRNA-miRNA interactions. Authors used in vitro based approaches to prove these interactions but these approaches does not authenticate the interactions that exist in vivo.

4. The authors point out that they have decided to study the network of CLDN4, but they are completely vague about the reason why they chose that particular transcript, or what were the criteria that drove their choice. Based on pathway analysis they have selected the following 4 most represented pathways: cell adhesion, pathway in cancer, cell cycle, and tight junction. While the first three are among the top candidates, the tight junction pathway stands at the bottom of the list of pathways identified for the upregulated genes. The p-val is 0.039 but with a high FDR (0.5). What was the cut-off chosen to proceed with a deeper analysis of the specific pathways? In addition, the authors claim that CLDN4 in the tight junction pathway drew their attention, but I cannot understand exactly why: is it outstanding for any reason? CLDN3 and CLDN7 are also part of the tight junction pathway and implicated in cancer progression and metastases, and show significant upregulation in the authors' analysis. Can the authors be more specific about their interest in this transcript?

5. Figure 1b: How was the network analysis performed? There are no details on how the interactions were retrieved. Within the pathway analysis, what do the authors mean with 'mRNA-ceRNA analysis was performed on choice classical genes...'.

6. Fig 3a. From the picture is very difficult to see a morphological difference between the three images, just a difference in the confluency. More importantly, figures and fonts are too small to go through and derive conclusions.

7. Fig3 e-j. Authors should compare their results with another set of gastric cancer samples to verify their claims (e.g. published larger datasets)? Has the specificity and sensitivity of K-M

analysis was verified using ROC analysis?

8. Fig 4 g-j, why have the authors used different cell lines for overexpression and knockdown experiments of lncRNA-KRTAP-AS1?

9. I found very strange sentences throughout the manuscript. The manuscript requires thorough revision and should be seen by a native English speaker.

Minor comments

Page 5, What is NAT?

Authors use term 'novel' for annotated and characterized transcripts.

What is ectopic miR-596 or miR-3620-3p transfection?

The following claim is wrong. Have the miRNAs shown to take part in these functions? The cited references explain only lncRNA functions but not miRNAs

Similar to miRNAs, lncRNAs can function as scaffolds or guides to regulate interaction between protein and genes, as decoys to bind proteins or miRNAs, and as enhancers to modulate transcription after transcribed from enhancer regions or their neighboring loci

Reviewer #2 (Remarks to the Author):

Review: Non-coding RNAs participate in the regulatory network of CLDN4 in a ceRNA-mediated miRNAs evasion

The authors report that they have discovered a network that is directed towards Claudin-4 (CLDN4) in gastric cancer (GC). They have observed that CLDN4 was up-regulated in GC and was associated with poor prognosis. CLDN4 (overexpression) reinforced proliferation, invasion, and EMT of GC cells, which could be reversed by transfection of exogenous miR-596, miR-3620-3p. Additionally, lncRNA-KRTAP5-AS1 and lncRNA-TUBB2A may act as competitively expressed (ce)RNA to affect the function of CLDN4. The authors suggest that non-coding RNAs (ncRNAs) play important roles in the regulatory network of CLDN4. As such, ncRNAs should be considered as a potential biomarker and therapeutic target against GC.

Comments

1. Recent studies report anew lncRNAs regulatory circuitry in which lncRNAs may function as competing endogenous RNAs (ceRNAs) and crosstalk with mRNAs by competitively binding their common miRNAs. The manuscript proposes such a model but the experiments to support the specifics of their model are lacking.

2. Title- "ce" is not readily evident to most readers. I would spell out competitively endogenous in the title.

3. Page 5- Studies were performed to correlate bioinformatic on genes, miRNAs and lncRNAs in GC tissue and neighboring non-tumor tissue. The authors compare 6 match-paired sets of tissue. It would be valuable to indicate whether the patients were newly diagnosed or treated. In addition, if treated, were they treated similarly or differently. Were they treated and in remission or in relapse?

4. The authors compare gene expression, miRNAs and lncRNAs using tissue from the same 6 patients. However, the bioinformatic results that are presented in Supp Table 2, 3, and 4 represent

the average of genes, miRNAs and lncRNAs that were up/downregulated. Since the authors are building a gene-miRNA-lncRNA network, values for each individual patient would be more informative. Also, how many replicates were performed for each patient based upon the GC tissue? Finally, the biology of GC, treatment modalities and therapeutic responses appear to differ with demographics. Please add further details on patient demographics and treatment details. First, the difference in the relative expression of CLDN4 in GC vs NAT is statistically but not biologically very significant. Second, the difference in expression of miR-596 and miR-3620-3p between GC and NAT is not dramatic.

5. Page 6- top. "According to pathway analysis results, we selected 4 classical pathways related to cancer development: "Cell adhesion", "Pathway in cancer", "Tight junction" and "Cell cycle". Combined with lncRNA and miRNA microarray results, mRNA-ceRNA analysis was performed on choice classical genes within these four pathways". How were the 4 pathways selected? Were the 4 pathways that were selected for study of GC biology selected based upon any metrics or based upon general interest? Similarly, how were the genes within each pathway selected for study? Related to this figure, Figure 1b is not informative.

6. Figure 1e- the loading control is unbalanced.

7. To test the biological function of CLDN4 and further verify its association with miR-596 and miR-3620-3p, the authors stably overexpressed CLDN4 in SGC-7901 cells. The gastric cancer cell line (SGC-7901) was established in our laboratory from the surgically resected metastatic lymph node of a female patient. It would be necessary to demonstrate comparable results in other GC cell lines. The reduction of proliferative capacity and invasion ability caused by knocking down CLDN4 could be largely rescued by inhibition of miR-596 or miR-3620-3p. Variants of miR-596 and miR-3620-3p would be valuable to show that the effect of miRNAs on CLDN4 and cell proliferation is specific.

8. In order to further investigate the role of CLDN4 in human GC, we performed deep validations in 62 pairs of GC tissues and NATs. Details on the patient demographics and treatment histories are lacking. With 82.26% of patients showing higher expression in cancer tissue, the transcript level of CLDN4 expression was significantly higher in the GC tissues compared with matched NATs. The median survival was 30 months (range, 5-81) in the higher CLDN4 group and 43 months (range, 2-81) in the lower CLDN4 group. The relative expression level of miR-596 and miR-3620-3p in human GC tissues compared with their matched NATs. The correlation between CLDN4 mRNA level and miR-596 or miR-3620-3p transcriptional levels were measured in the same set of patients by Spearman correlation analysis. In Figure 3, it would be valuable to graph panels e, g, and h together to determine whether CLDN4 expression correlates with miR expression of each patient.

9. In Figure 4, direct binding experiments are performed to demonstrate lncRNA crosstalk with miRNAs. These experiments should first be performed in multiple different GC cell lines. Second, they need to be performed using miR and lnc mutant variants of the endogenous miR and lnc to demonstrate specificity.

10. To evaluate the biological functions of these genes *in vivo*, different SGC-7901 cells were subcutaneously injected into nude mice. The authors report that CLDN4 promotes SGC-7901 growth in orthotopic transplants. The authors that report that the effect of CLDN4 can be inhibited by miR-596, miR-3620-3p and enhanced by lncRNA-KRTAP5-AS1, lncRNA-TUBB2A. However, these effects are on the inhibition of tumor growth. The authors should demonstrate that, after tumors are allowed to form, that expression of CLDN4, miR-596 and miR-3620-3p modulate tumor growth.

11. The discussion and model require further support from the experiments suggested above.

Reviewer #3 (Remarks to the Author):

In the manuscript "Non-coding RNAs participate in the regulatory network of CLDN4 in a ceRNA-mediated miRNAs evasion" the authors claim that they have discovered CLDN4 as part of a network important for gastric cancer (GC) development. They state that CLDN4 is up-regulated in GC and associated with poor prognosis. They perform extensive in vitro and in vivo experimental work and based on this claim that CLDN4 reinforced proliferation, invasion, and EMT of GC cells, which could be reversed by miR-596, miR-3620-3p. They also claim that the lncRNAs KRTAP5-AS1 and TUBB2A act as ceRNAs for miR-596 and miR-3620-3p and through this affect CLDN4. They claim that CLDN4 is a potential biomarker and therapeutic target against GC.

The major weaknesses of the manuscript are:

1) The bioinformatic (GO and pathway analyses) analyses that led to the focus on CLDN4 is poorly described and based on very small clinical sample numbers. Accordingly, it is not clear to me how the authors ended up analyzing CLDN4.

2) the data supporting a clinical relevance of CLDN4 is weak.

Based on analysis of 62 GC patients the authors state that the CLDN4 upregulation is associated with poor prognosis after gastrectomy. However the authors do not describe how they set the threshold for dividing the CLDN4 expression into low and high. The observed survival difference is very small and likely highly dependent on the defined threshold.

Is the expression levels of the miRs also prognostic? could it be them (independently of CLDN4 that are cause of any survival difference. The authors do not investigate if the CLDN4 observation could be confounded. Is the effect of CLDN4 independent of tumor stage (and other clinical variables)?

To increase confidence that CLDN4 expression is associated with prognosis. The finding should be validated in an independent clinical cohort.

Using the same cut-off used here

3) The authors state that CLDN4 and miR-596, miR-3620-3p are inversely expressed in GC tissues.

However, the data supporting this statement are not very solid.

It is not surprising that correlation analysis of an upregulated and a downregulated transcript may show some correlation, but this does not mean causality. There may very well be no direct interaction between the two in tissues. The authors will most likely find that the miRs show a weak negative correlation to the majority of the genes they find upregulated. This will in most cases be due to non-causal coincidence.

The authors report correlation coefficients of 0.323 and 0.356, indicating that at best 10-12% of the variation in CLDN4 can be explained by the miRNAs.

This practically nothing.

I wonder if the miRNAs show any correlation to CLDN4 when the tumors and NATs are analyzed separately? This will eliminate the bias that they are comparing opposite deregulated. If CLDN4 and the miRs truly interact they should be negatively correlated also when tumor and NATs are analyzed separately.

I wonder if there is matched mRNA and miRNA public available on clinical GC samples that can be used to validate the findings...maybe from the TCGA ???

4) That the in vitro and in vivo results are based on two cell lines only. CLDN4 is over-expressed in SGC-7901 cells and knock down is performed in AGS cells. Can the findings be replicated in other GC cell lines or are they peculiarities of the selected lines?

The xenograft experiments were also made with the sample cell lines. Without replication in additional cell lines it is impossible to assess if the findings have any general value.

The authors should examine a range of GC cell lines to document that the findings are general.

5) There appears to be no analysis of the expression of the lncRNAs KRTAP5-AS1 and TUBB2A in the 62 clinical samples. How are their expression levels correlated with CLDN4 in the clinical samples?

6) There are no experiments showing which cells actually express CLDN4, the miRNAs and the lncRNAs in the clinical samples!!! The literature contains numerous examples that miRNAs apparently deregulated in epithelial tumors, where in fact expressed by stromal cells and not epithelial cells, and the deregulation was merely a consequence of altered tissue composition. In these situations it does not make sense to overexpress the miRNAs in epithelial cells. The authors should investigate which cell types in the clinical samples do express the essential genes of the CLDN4 network.

Minor comments

1) Many of the figures were small to be properly assessed Fig 1A, Fig 2, Fig 4 and Fig 5.

2) There is no clinical info provided on the 62 GC cases, only on the 6 used for microarray analysis.

3) The PSiCheck2-CLDN4 vector is not mentioned in the mat and met section. There is no description of what part of CLDN4 that was cloned into the vector making it impossible to assess the validity of the luc-results in fig 1.

While the Psicheck2-TUBB2A and Psicheck2-KRTAP5-AS1 vectors are mentioned, there is again no description of what was cloned.

4) the mRNA-ceRNA analysis is not described in any detail anywhere

5) How was the transcripts shown in figure 1A selected? The figure appears to not to contain all deregulated transcripts ? According to page 5 there were several thousand deregulated mRNAs and lncRNAs and a few hundred deregulated miRNAs

6) The language is poor many places throughout the manuscript, though particularly in the discussion. Here are a number of examples

From page 9 "With 82.26% of patients showing higher expression in cancer tissue, the transcript level of CLDN4 expression was significant higher in the GC tissues compared with matched NATs via Wilcoxon's..."

The following examples are all from the discussion

"Although there is still some controversy on the role of CLDN4 on carcinogenesis 45, many studies have revealed CLDN4's oncogenic functions such as enhancing proliferation, invasion and EMT"

"The oncogenic role of CLDN4 in enhancing invasion and EMT can be explained by that claudin family members could activate the MMP family members and ZEB family members, as showed by previous studies"

"These strengthening effects of CLDN4 are weakened after the exogenous introduction of miR -596 and miR-3620-3p."

"As it is well known that a single miRNA can bind multiple targeting mRNAs through miRNA response elements (MREs)"

"This can be explained that lncRNAs, which possess the capacity to function as ceRNAs, can carry target MREs for various mRNA-targeting miRNAs, resembling the fact that miRNAs can target multiple mRNAs"

Response to Reviewer #1

1. The major concern of the current work is the specificity of the regulatory interactions. For example, miR-596 and miR-3620-3p can induce apoptosis and cell cycle perturbations independent of the CLDN4 activity. LncRNAs also perform functions independent of CLDN4. This indicates that partial rescue of CLDN4 mediated functions could be due to their influence on the pathways that are independently controlled by miRNAs and lncRNAs. Likewise, can the authors claim that the effects on invasion and proliferation are fully due to the direct interaction of CLDN4 with these two miRNAs, and not as result of combined effect on other targets with the same function? For example, these 2 miRNAs are also predicted to target CLDN7, a gene product that has been identified by the authors as upregulated in gastric cancer and also part of the tight junction network.

Thank you for your excellent question. In order to prove the specificity of the regulatory interactions, we knocked out *CLDN4* in AGS cells and monitored the change in their invasive and proliferative capacities after separately transfecting them with miR-596, miR-3620-3p, lncRNA-KRTAP5-AS1, or lncRNA-TUBB2A. The results showed a significant decrease in invasion and proliferation ability after knocking down *CLDN4* in AGS cells compared with the control group, as expected. However, we did not find an obvious change after separately transfecting the two miRNAs and lncRNAs into *CLDN4* knockdown AGS cells. These results indicate that the miRNAs and lncRNAs could not function without *CLDN4* present. Thus, combining these new results with our previous findings, we believe that the effects on invasion and proliferation are fully due to the direct interaction of these two miRNAs and lncRNAs with *CLDN4*. Details of these additional experiments have been added in Supplementary Fig.3c-3d and shown below:

Figure legend: (a) Cell proliferation was assessed daily for four days using the Cell Counting Kit-8 (CCK-8) assay in *CLDN4* knockdown AGS cells. (b) Transwell assays were performed in *CLDN4* knockdown AGS cells. Cells were incubated for 24h and counted under the microscope. Original magnification $\times 200$. Scale bars = $100\mu\text{m}$. Data are shown as mean \pm SD, $n=3$. *represents $P < 0.05$.

2. The authors should try to verify a synergic effect of the 2 miRNAs on *CLDN4* expression and function if transfected simultaneously?

We appreciate your suggestion to further explore the effects of the miRNAs. We have transfected mimics of miR-596 and miR-3620-3p simultaneously or individually into gastric cancer cells. The total dose of the co-transfection was $5\mu\text{l}$ ($2.5\mu\text{l}$ miR-596 and $2.5\mu\text{l}$ miR-3620-3p), and the total dose of a single transfection was $5\mu\text{l}$. We saw that the two miRNAs transfected simultaneously could significantly inhibit *CLDN4*

expression and functions. The effect of the two miRNAs transfected together was similar to each one individually when transfected as a double dose. Thus, it appears that they have additive, not synergistic, effects on CLDN4. All of the above results are shown as followed:

Figure legend: (a and b) The relative expression levels of CLDN4 were determined by real-time PCR and western blotting in GC cells treated with mimics of miR-596, miR-3620-3p individually

and simultaneously. (c) Cell proliferation was assessed daily for four days using CCK-8 assay in CLDN4 overexpressing SGC-7901 cells. (d) Transwell assays were performed in CLDN4 overexpressing SGC-7901 cells. (e) Cell proliferation was assessed daily for four days by CCK-8 assay in CLDN4 overexpressing AGS cells. (f) Transwell assays were performed in CLDN4 overexpressing AGS cells. (g) Cell proliferation was assessed daily for four days by CCK-8 assay in CLDN4 overexpressing HGC-27 cells. (h) Transwell assays were performed in CLDN4 overexpressing HGC-27 cells. In (d, f and h): Cells were incubated for 24h and counted under the microscope. Original magnification×200. Scale bars=100µm. Data are shown as mean±SD, n=3. *represents P<0.05.

3. It would be better if authors provide additional data to reinforce the specificity of these regulatory interactions. By performing RNA-seq, authors can characterize molecular pathways that are controlled by miRNA, lncRNA and CLDN4 in gastric cancer cell lines. This unbiased strategy will enable us to know whether these RNAs regulate common pathways.

This is a good suggestion. We performed RNA-sequencing on six gastric cancer cell lines (SGC-7901 cells overexpressing miR-596, overexpressing miR-3620-3p, miRNA negative control, overexpressing lncRNA-KRTAP5-AS1, overexpressing lncRNA-TUBB2A, or lncRNA negative control). The process of RNA-sequencing was added to Methods. Our results reinforced the specificity of the regulatory interactions. In miR-596 overexpressing cells, the expression of CLDN4 (fold-change of FPKM=0.125) and lncRNA-KRTAP5-AS1 (fold-change of FPKM=0.760) were down-regulated. In addition, the expression of CLDN4 (fold-change of FPKM=0.543), lncRNA-TUBB2A (fold-change of FPKM=0.854), and lncRNA-KRTAP5-AS1 (fold-change of FPKM=0.280) were also down-regulated in cells overexpressing miR-3620-3p compared with the control group. Interestingly, increased expression of CLDN4 was also found in cells overexpressing lncRNA-KRTAP5-AS1 (fold-change of FPKM=14.168) or lncRNA-TUBB2A (fold-change of FPKM=12.051). We have added the contents above and the original data of the RNA-sequencing to Results, Methods, and Supplementary Table 11.

Furthermore, we performed pathway analysis based on following methods:

Gene type	The process of pathway analysis
miRNAs (miR-596 or	(1) Prediction of target mRNAs using TargetScan (Release 7.1)

miR-3620-3p	(2) Filter predictive target mRNAs with a cutoff of fold-change<0.5 between the overexpressed miRNA group and the miRNA control group (3) KEGG pathway analysis using clusterProfiler package (v2.2.7) [Yu G et al. OMICS. 2012, 16(5):284-7]
lncRNAs (lncRNA-KRTAP5-AS1 or lncRNA-TUBB2A)	(1) Cis or Trans Regulating mRNA Prediction of lncRNA: a) Cis regulation - coding genes within 300kb on the same chromosome b) Trans regulation – LncTar [Li J et al. Brief bioinform. 2015, 16(5):806-812] software with a cutoff of ndG<-0.1 (2) Filter the predictive target mRNAs based on fold-change<0.5 or fold-change>2 between the overexpressed lncRNA group and the lncRNA control group (3) KEGG pathway analysis using clusterProfiler package (v2.2.7) [Yu G et al. OMICS. 2012, 16(5):284-7]
mRNA (CLDN4)	Obtain pathways from: http://www.kegg.jp/kegg/pathway.html

The resulting common pathways regulated by these RNAs were as listed:

Pathways	CLDN4	miR-596	miR-3620-3p	lncRNA-KRTAP5-AS1	lncRNA-TUBB2A
Tight junction	+	-	-	-	+(P=0.038)
Cell adhesion molecules (CAMs)	+	-	-	+(P=0.007)	-
Hepatitis C	+	-	-	+(P=0.024)	-
Leukocyte transendothelial migration	+	-	-	+(P=0.015)	-
cAMP signaling pathway	-	+(P=0.021)	+(P=0.022)	-	-
FoxO signaling pathway	-	+(P=0.034)	+(P=0.039)	-	-
Inflammatory mediator regulation of TRP channels	-	+(P=0.015)	+(P=0.048)	-	-
Apoptosis	-	+(P=0.006)	-	+(P=0.029)	-
MAPK signaling pathway	-	+(P=0.048)	-	+(P=0.005)	+(P=0.003)
Cytokine-cytokine receptor interaction	-	-	+(P=0.037)	+(P=0.002)	-
Ras signaling pathway	-	-	-	+(P=0.027)	+(P=0.012)

Authors should also provide in vivo evidence on mRNA-miRNA and lncRNA-miRNA interactions. Authors used in vitro based approaches to prove these interactions but these approaches does not authenticate the interactions that exist in vivo.

In the revised manuscript, we have provided in vivo evidence on mRNA-miRNA and lncRNA-miRNA interactions. We subcutaneously injected the same amount of untreated SGC-7901 cells into the both sides of nude mice. After tumors formed, we injected agomir of miRNA-596 or miR-3620-3p into the tumors of one side and negative controls into the other side for five total injections. The tumors injected with

miR-596 or miR-3620-3p agomir were smaller than the ones injected with negative control (See Fig.a-b below). Furthermore, the tumors injected with miR-596 agomir showed less CLDN4 expression on both the translational and transcriptional levels, as well as less lncRNA-KRTAP5-AS1 at the transcriptional level. Similarly, the tumors injected with miR-3620-3p agomir showed less CLDN4, lncRNA-TUBB2A, and lncRNA-KRTAP5-AS1 expression (See Fig.c-f below). We added all the new results in Fig.8 of the revised manuscript.

*Figure legend: SGC-7901 cells were subcutaneously injected at both right and left armpit region of nude mice. After tumor formation, agomirs of miRNAs were injected into tumors of right side and their negative controls into left side. (a and b) Treatment of miR-596 and miR-3620-3p suppressed xenograft tumor growth. (c and d) miR-596 and miR-3620-3p treated tumors showed less CLDN4 expression. (e) miR-596 and miR-3620-3p treated tumors showed less lncRNA-KRTAP5-AS1 expression. (f) miR-3620-3p treated tumors showed less lncRNA-TUBB2A expression. Data are mean±SD, * means P<0.05, ** means P<0.01.*

4. The authors point out that they have decided to study the network of CLDN4, but they are completely vague about the reason why they chose that particular transcript, or what were the criteria that drove their choice. Based on pathway analysis they have selected the following 4 most represented pathways: cell adhesion, pathway in cancer,

cell cycle, and tight junction. While the first three are among the top candidates, the tight junction pathway stands at the bottom of the list of pathways identified for the upregulated genes. The p-val is 0.039 but with a high FDR (0.5). What was the cut-off chosen to proceed with a deeper analysis of the specific pathways? In addition, the authors claim that CLDN4 in the tight junction pathway drew their attention, but I cannot understand exactly why: is it outstanding for any reason? CLDN3 and CLDN7 are also part of the tight junction pathway and implicated in cancer progression and metastases, and show significant upregulation in the authors' analysis. Can the authors be more specific about their interest in this transcript?

We appreciate your suggestion to expound upon why we chose CLDN4. In our study, we chose the cut-off based on the p-val. All pathways with a p-val less than 0.05 were proceeded with a deeper analysis. Considering the small number of samples used for the microarray, the FDR values were usually high and not suitable for estimating the cut-off. In addition, the tight junction pathway caught our attention since our previous work demonstrated aberrant expression of CLDN4, a component of the tight junction pathway, in gastric cancer and precursor lesions [Zhu et al. World J Surg Oncol.2013,11(1):150]. Also, using meta-analysis, we have previously found that CLDN4 expression was associated with increasing pT category, tumor size, and lymph node metastasis in patients with gastric cancer [Chen et al. Onco Targets Ther. 2016,9:3205-12]. Therefore, we focused on the tight junction pathway and chose CLDN4 for our key focus point. Following your suggestion, we have added the above content to Introduction.

5. Figure 1b: How was the network analysis performed? There are no details on how the interactions were retrieved. Within the pathway analysis, what do the authors mean with 'mRNA-ceRNA analysis was performed on choice classical genes...'.

We agree with the reviewer that we should clarify these processes. The details of the network analysis and mRNA-ceRNA analysis are as listed:

i) Six pairs of GC tissues and non-tumorous adjacent tissues were analyzed via microarray using the Human LncRNA + mRNA Array v2.0 together with the miRCURY LNATM microRNA Array.

ii) Using a Student's t-test, we identified 4421 differentially expressed lncRNAs and 3369 differentially expressed mRNAs, all of which were up-regulated or down-regulated more than 2 fold (Supplementary Table 6-7).

iii) Based on the KEGG database, we performed pathway analysis for differentially expressed mRNAs. We chose the cut-off based on p-val, and all pathways with a p-val less than 0.05 proceeded with a deeper analysis. We identified 22 pathways based on the up-regulated genes and 39 pathways based on the down-regulated genes.

iv) Considering the p-val and our previous work, the 61 pathways were filtered down to 4 pathways of interest. Cell adhesion, pathway in cancer, and cell cycle were among the top candidates and related to cancer development. Moreover, we selected the tight junction pathway because our previous work demonstrated aberrant expression of CLDN4, a component of the tight junction pathway, in gastric cancer and precursor lesions [Zhu et al. *World J Surg Oncol.* 2013,11(1):150]. Also, using meta-analysis, we previously found that CLDN4 expression was associated with increasing pT category, tumor size, and lymph node metastasis in patients with gastric cancer [Chen et al. *Onco Targets Ther.* 2016,9:3205-12]. Therefore, we focused on the tight junction pathway and chose CLDN4 as our key target.

v) Following the pathway selection, we chose key mRNAs from the differentially expressed mRNA list for each pathway according to previous reports on their relation with gastric cancer and their p-val. For example, there were 20 significantly highly-expressed mRNAs in the tight junction pathway, and 4 of them (CLDN1, CLDN4, CLDN7 and HRAS) were reported to be highly expressed in gastric cancer. Considering that the p-val of HRAS was relatively high ($p=0.019$), we only selected CLDN1, CLDN4, and CLDN7 to proceed with a deeper analysis. After this selection, the remaining key genes were:

Pathways	Selected key genes
Cell adhesion	CLDN18; ITGA4; PTPRC
Pathway in cancer	RAC1; VEGFA; MMP1; ERBB2; HRAS; IL8
Tight junction	CLDN4; CLDN7; CLDN1
Cell cycle	CCNB1; CCNE1; CDK1; CDK7; SFN; MYC

vi) We then predicted the miRNAs which could regulate the key mRNAs above. There were three criteria: the expression of a miRNA was significantly different between cancer and non-tumorous adjacent tissues; the miRNA was up-regulated if the candidate target mRNA was down-regulated, or down-regulated if the candidate target mRNA was up-regulated; and context score<0 [Grimson et al. Mol Cell. 2007,27(1):91-105] and context+ score<0 [Friedman et al. Genome Res. 2009,19(1):92-105; Garcia et al. Nat Struct Mol Biol. 2011,18(10):1139-46].

vii) After removing the mRNAs with no apparent miRNA regulation, the list of miRNA-mRNA interactions was as follows:

Pathways	miRNA-mRNA pairs
Cell adhesion	hsa-miR-3689b-3p & CLDN18;
	hsa-miR-3689c & CLDN18
	hsa-miR-3121-5p & ITGA4
Pathway in cancer	hsa-miR-22-5p & MMP1
	hsa-miR-4292 & HRAS
	hsa-miR-136-5p & RAC1
	hsa-miR-1470 & VEGFA
Tight junction	hsa-miR-125b-5p & VEGFA
	hsa-miR-596 & CLDN4
	hsa-miR-3620-3p & CLDN4
	hsa-miR-4292 & CLDN4
	hsa-miR-624-5p & CLDN1
	hsa-miR-596 & CLDN7
Cell cycle	hsa-miR-3620-3p & CLDN7
	hsa-miR-628-3p & CDK1
	hsa-miR-22-5p & CDK1

viii) Potential ceRNAs were predicted for each miRNA-mRNA pair using the MuTaMe Score [Tay et al. Cell. 2011,147(2):344-57], with the cut-off set as MuTaMe Score>0.005. All 4421 differentially expressed lncRNAs and 3369 differentially expressed mRNAs were entered into this ceRNA prediction as candidates (Supplementary Table 6-7). The interactions were graphed using the Cytoscape software (v2.8.1) (Fig.1b).

We added the process of our network analysis and mRNA-ceRNA analysis to Supplementary Methods and we revised Results to include: “Details regarding the

process of pathway analysis, mRNA-ceRNA analysis, and selection of key genes are listed in the Supplementary Methods.”

6. Fig 3a. From the picture is very difficult to see a morphological difference between the three images, just a difference in the confluency. More importantly, figures and fonts are too small to go through and derive conclusions.

Thank you for pointing this out. We have now provided new representative images and have edited the figures and fonts to make them larger and more clear (See Figure below). The new pictures are shown in Fig.3a of the revised manuscript.

Figure legend: Phase-contrast micrographs of CLDN4 overexpressing cells, pEX2-NC cells and SGC-7901 cells. Scale bars=50µm.

7. Fig3 e-j. Authors should compare their results with another set of gastric cancer samples to verify their claims (e.g. published larger datasets)? Has the specificity and sensitivity of K-M analysis was verified using ROC analysis?

Thank you for your excellent comments. According to your suggestion, we measured the expression of CLDN4 in the TCGA database. We found that the RPKM of CLDN4 was significantly higher in 29 GC tissues compared with their matched non-tumorous adjacent tissues ($P < 0.001$, See Fig.a below). This result was in accordance with our earlier finding. However, only five pairs of available data could be used for calculating the expression of miR-3620-3p, and no pairs were found that could be used for miR-596. Thus, due to the small sample size, we could not analyze the expression of miRNAs in the TCGA database. Instead, we measured the expression of CLDN4 and miRNAs in an additional independent cohort of 42 GC patients. The results confirmed that miR-596 and miR-3620-3p expression were both significantly lower in the GC tissues ($P < 0.05$ for both, See Fig.b-c below). Significantly higher CLDN4 expression could also be detected in the GC tissues and was associated with poor prognosis ($P = 0.011$, See Fig.d below).

According to your suggestion, we performed ROC analysis among all 104 GC patients (including the original 62 patients and the additional cohort of 42 patients). The results verified the specificity and sensitivity of the K-M analysis ($P=0.001$ for K-M analysis, $P=0.007$ for ROC analysis, See Fig.e-f below). The Youden index was maximized when the cut-off value of CLDN4 was set as 2.343 fold, which was equal to the cut-off used for the K-M analysis in the original submission of this manuscript. When using this cut-off, the sensitivity and specificity were 65.1 % and 70.7 %, respectively.

Figure legend: (a) The RPKM of CLDN4 in 29 GC tissues and their matched non-tumorous adjacent tissues from TCGA database. (b and c) The value of ΔCT (miRNAs normalized to U6 snRNA) was used to compare the relative expression of miR-596 and miR-3620-3p in GC tissues and matched non-tumorous adjacent tissues among the additional cohort of 42 GC patients. Larger ΔCT value indicated lower expression. (d) Kaplan-Meier analysis of the correlation between overall survival and CLDN4 expression level in the additional cohort of 42 GC patients. (e) Kaplan-Meier analysis of the correlation between overall survival and CLDN4 expression level in all 104 patients. (f) The corresponding ROC curve based on the Kaplan-Meier analysis performed in all 104 patients.

8. Fig 4 g-j, why have the authors used different cell lines for overexpression and knockdown experiments of lncRNA-KRTAP-AS1?

Following your suggestion, we repeated the overexpression and knockdown experiments in the SGC-7901 cell line, the AGS cell line, and the HGC-27 cell line.

We have added the data from the overexpression and knockdown experiments for lncRNA-KRTAP5-AS1 in each cell line to Fig.5a-5h and Supplementary Fig.8c-8f. All of the results in the multiple cell lines had a common trend showing that lncRNA-KRTAP5-AS1 could promote proliferation, invasion, and EMT as a ceRNA in gastric cancer. We have included text in Results detailing the above results in the revised manuscript.

9. I found very strange sentences throughout the manuscript. The manuscript requires thorough revision and should be seen by a native English speaker.

Thank you for pointing this out. We have collaborated with a native English speaking biologist to correct all of the grammatical errors and edit the language of the whole manuscript.

Minor comments

1. Page 5, What is NAT?

NAT is the abbreviation of non-tumorous adjacent tissues. We have changed all “NAT” to “non-tumorous adjacent tissues” in the manuscript.

2. Authors use term ‘novel’ for annotated and characterized transcripts.

It has been removed.

3. What is ectopic miR-596 or miR-3620-3p transfection?

We changed it to “This CLDN4 up-regulation caused by stable transfection could be overcome at both the transcriptional and translational levels by ectopic overexpression of miR-596 or miR-3620-3p”

4. The following claim is wrong. Have the miRNAs shown to take part in these functions? The cited references explain only lncRNA functions but not miRNAs. Similar to miRNAs, lncRNAs can function as scaffolds or guides to regulate interaction between protein and genes, as decoys to bind proteins or miRNAs, and as enhancers to modulate transcription after transcribed from enhancer regions or their neighboring loci

We changed it to “LncRNAs are now known to have many functions, acting as scaffolds or guides to regulate interactions between protein and genes, as decoys to

bind proteins or miRNAs, and as enhancers to modulate transcription of their targets after being transcribed from enhancer regions or their neighboring loci.”

Response to Reviewer #2

1. Recent studies report a new lncRNAs regulatory circuitry in which lncRNAs may function as competing endogenous RNAs (ceRNAs) and crosstalk with mRNAs by competitively binding their common miRNAs. The manuscript proposes such a model but the experiments to support the specifics of their model are lacking.

Thank you for your suggestion. Using RNA immunoprecipitation (RIP), RNA pull-down, and luciferase reporter assays, we verified that lncRNA-KRTAP5-AS1 can physically interact with miR-596 and miR-3620-3p and that lncRNA-TUBB2A can interact with miR-3620-3p. In response to your suggestion, we performed RIP assays based on Ago2, which can enrich for targets bound by miRNAs upon immunoprecipitation. We separately overexpressed lncRNA-KRTAP5-AS1 and lncRNA-TUBB2A in SGC-7901 and AGS cells, then used an anti-Ago2 antibody to pull down Ago2. Overexpression of lncRNA-KRTAP5-AS1 or lncRNA-TUBB2A both caused a significant decrease in the enrichment of CLDN4 transcripts pulled down by Ago2, indicating that there were less miRNA-bound CLDN4 transcripts present (See Fig.a below). This suggests that lncRNA-KRTAP5-AS1 and lncRNA-TUBB2A can compete with the CLDN4 transcript for the binding of miRNAs. Additionally, we performed RIP experiments and RNA pull-down assays in the AGS and SGC-7901 cell lines using lncRNAs in which their binding sequences for binding to miRNA were artificially mutated. The results of the RIP and RNA pull-down showed that lncRNA-KRTAP5-AS1 could bind miR-596 and miR-3620-3p while the lncRNA-KRTAP5-AS1 mutant or empty vector (MS2) have no interaction with miRNAs. Similarly, lncRNA-TUBB2A could bind miR-3620-3p, but the lncRNA-TUBB2A mutant could not (See Fig.b-c below). We added all of these new results in Fig.4d-4e and Fig.6b of the revised manuscript.

Figure legend: (a) The schematic diagram and real-time PCR results of the RIP based on Ago2 showed that lncRNAs can compete with the CLDN4 transcript for the binding of miRNAs. (b) The schematic diagram and real-time PCR results of the MS2-RIP method used to identify the binding between lncRNAs and miRNAs in both SGC-7901 and AGS cells. (c) The schematic diagram of the RNA pull down method used to identify the binding between lncRNAs and miRNAs in both SGC-7901 and AGS cells. GC cell lysates were incubated with biotin-labeled lncRNA-KRTAP5-AS1, lncRNA-TUBB2A, lncRNA-KRTAP5-AS1-mut, and lncRNA-TUBB2A-mut. MiRNA real-time PCR was performed after pull down process. Data are shown as mean±SD, n=3. *represent P<0.05, **represent P<0.01.

2. Title- “ce” is not readily evident to most readers. I would spell out competitively endogenous in the title.

We changed it to “competing endogenous RNA” in the title.

3. Page 5- Studies were performed to correlate bioinformatic on genes, miRNAs and lncRNAs in GC tissue and neighboring non-tumor tissue. The authors compare 6 match-paired sets of tissue. it would be valuable to indicate whether the patients were newly diagnosed or treated. In addition, if treated, were they treated similarly or differently. were they treated and in remission or in relapse?

Thank you for your suggestion. All six GC patients were newly diagnosed, and they did not receive any therapy. According to your suggestions, we added details for the

six match-paired sets of tissues in Methods as follows: “Six match-paired sets of tissues for microarray were obtained from patients who were newly diagnosed with GC and received radical resection at the First Affiliated Hospital of China Medical University.” Moreover, we also updated the clinical information for these six patients in Supplementary Table 1.

4. The authors compare gene expression, miRNAs and lncRNAs using tissue from the same 6 patients. However, the bioinformatic results that are presented in Supp Table 2, 3, and 4 represent the average of genes, miRNAs and lncRNAs that were up/downregulated. Since the authors are building a gene-miRNA-lncRNA network, values for each individual patient would be more informative. Also, how many replicates were performed for each patient based upon the GC tissue?

We appreciate this kind suggestion very much. According to your suggestion, individual values for each patient were uploaded as Supplementary Tables 2-4.

There were no replicates performed for each individual patient due to the scarcity of GC tissue. However, in the microarray analysis, the six match-paired sets of tissue represented biological replicates. According to tests for sample quality and variation, our microarray is reliable: i) for optimal array hybridization, it is normally recommended to use abs260/abs280 ratios above 1.8 for good array performance. All abs260/abs280 ratios for our samples were above 1.99, representing good sample quality. ii) The Pearson correlation analysis was used to assess the reproducibility between chips. All of the correlation coefficients for our tests were above 0.5 and most of the correlation coefficients were above 0.9, representing good reproducibility. Furthermore, a series of experiments (real-time PCR, western blotting, RIP) further confirmed the bioinformatics results on the gene-miRNA-lncRNA network.

In addition, we performed RNA-sequencing on six gastric cancer cell lines (SGC-7901 cells overexpressing miR-3620-3p, overexpressing miR-596, miRNA negative control, overexpressing lncRNA-KRTAP5-AS1, overexpressing lncRNA-TUBB2A, or lncRNA negative control). In cells overexpressing miR-596, the expression of CLDN4 (fold-change of FPKM=0.125) and lncRNA-KRTAP5-AS1 (fold-change of FPKM=0.760) were down-regulated. In addition, the expression of

CLDN4 (fold-change of FPKM=0.543), *lncRNA-TUBB2A* (fold-change of FPKM=0.854), and *lncRNA-KRTAP5-AS1* (fold-change of FPKM=0.280) were down-regulated in cells overexpressing miR-3620-3p compared with the control group. Interestingly, increased expression of *CLDN4* was found in cells overexpressing *lncRNA-KRTAP5-AS1* (fold-change of FPKM=14.168) or *lncRNA-TUBB2A* (fold-change of FPKM=12.051). In conclusion, the RNA-sequencing verified the results of our microarray. We have added the contents above and original data from the RNA-sequencing to the Results, Methods, and Supplementary Table 11.

Finally, the biology of GC, treatment modalities and therapeutic responses appear to differ with demographics. Please add further details on patient demographics and treatment details. First, the difference in the relative expression of *CLDN4* in GC vs NAT is statistically but not biologically very significant. Second, the difference in expression of miR-596 and miR-3620-3p between GC and NAT is not dramatic.

According to your suggestion, we provided detailed information on the demographics, clinicopathological data, and treatment for each patient (Supplementary Table 1). In consideration of the small sample size, less stability and accuracy in microarray detection, we measured CLDN4 expression in the original 62 GC patients as well as in an additional cohort of 42 GC patients to confirm the microarray results. These results further confirmed that CLDN4 expression is significantly higher in GC tissues compared with matched non-tumorous adjacent tissues ($P < 0.001$, Fig.3e, Supplementary Fig.4e).

On the other hand, the initial microarray results indicated that the difference in expression of miR-596 and miR-3620-3p between GC and normal adjacent tissues is statistically significant but not dramatic ($P = 0.037$ for miR-596, $P = 0.036$ for miR-3620-3p). But these results were based on only six patients with GC. In order to confirm the expression of miR-596 and miR-3620-3p in GC, we measured their expression in the above 104 pairs of GC tissues. The expanded results confirmed that miR-596 and miR-3620-3p expression were both significantly lower in the GC tissues,

with a much more dramatic difference ($P < 0.001$ for both, Fig.3g-3h, Supplementary Fig.4e).

5. Page 6-top. “According to pathway analysis results, we selected 4 classical pathways related to cancer development: “Cell adhesion”, “Pathway in cancer”, “Tight junction” and “Cell cycle”. Combined with lncRNA and miRNA microarray results, mRNA-ceRNA analysis was performed on choice classical genes within these four pathways“. How were the 4 pathways selected? Were the 4 pathways that were selected for study of GC biology selected based upon any metrics or based upon general interest? Similarly, how were the genes within each pathway selected for study? Related to this figure, Figure 1b is not informative.

We appreciate your suggestion very much. To make Figure 1b more informative, we added the process of pathway and key gene selection to the Supplementary Methods.

Pathway selection:

i) Six pairs of GC tissues and non-tumorous adjacent tissues were analyzed via microarray using the Human LncRNA + mRNA Array v2.0 together with the miRCURY LNATM microRNA Array.

ii) Using a Student's t-test, we identified 4421 differentially expressed lncRNAs and 3369 differentially expressed mRNAs which were up-regulated or down-regulated more than 2 fold (Supplementary Table 6-7).

iii) Based on the KEGG database, we performed pathway analysis for differentially expressed mRNAs. We chose the cut-off based on p-val and all pathways with a p-val less than 0.05 proceeded with a deeper analysis. We identified 22 pathways based on the up-regulated genes and 39 pathways based on the down-regulated genes.

iv) Considering the p-val and our previous work, the 61 pathways were filtered down to four pathways of interest. Cell adhesion, pathway in cancer, and cell cycle were among the top candidates and all related to cancer development. Moreover, we selected the tight junction pathway because our previous work demonstrated aberrant expression of CLDN4, a component of the tight junction pathway, in gastric cancer and precursor lesions [Zhu et al. World J Surg Oncol. 2013,11(1):150]. Also, using meta-analysis, we previously found that CLDN4 expression was associated with

increasing pT category, tumor size, and lymph node metastasis in patients with gastric cancer [Chen et al. *Onco Targets Ther.* 2016,9:3205-12]. Therefore, we focused on the tight junction pathway and chose CLDN4 as our key target.

Key genes selection:

i) Following the pathway selection, we chose key mRNAs from the differentially expressed mRNA list for each pathway according to previous reports on their relation with gastric cancer and their p-val. For example, there were 20 significantly highly-expressed mRNAs in the tight junction pathway, and 4 of them (CLDN1, CLDN4, CLDN7 and HRAS) were reported to be highly expressed in gastric cancer. Considering that the p-val of HRAS was relatively high (p=0.019), we only selected CLDN1, CLDN4, and CLDN7 to proceed with a deeper analysis. After this selection, the remaining key genes were:

Pathways	Selected key genes
Cell adhesion	CLDN18; ITGA4; PTPRC
Pathway in cancer	RAC1; VEGFA; MMP1; ERBB2; HRAS; IL8
Tight junction	CLDN4; CLDN7; CLDN1
Cell cycle	CCNB1; CCNE1; CDK1; CDK7; SFN; MYC

ii) We then predicted the miRNAs which could regulate the key mRNAs above. There were three criteria: the expression of a miRNA was significantly different between cancer and non-tumorous adjacent tissues; the miRNA was up-regulated if the candidate target mRNA was down-regulated, or down-regulated if the candidate target mRNA was up-regulated; and context score<0 [Grimson et al. *Mol Cell.* 2007,27(1):91-105] and context+ score<0 [Friedman et al. *Genome Res.* 2009,19(1):92-105; Garcia et al. *Nat Struct Mol Biol.* 2011,18(10):1139-46].

iii) After removing the mRNAs with no apparent miRNA regulation, the list of miRNA-mRNA interactions was as follows:

Pathways	MiRNA-mRNA pairs
Cell adhesion	hsa-miR-3689b-3p & CLDN18;
	hsa-miR-3689c & CLDN18
	hsa-miR-3121-5p & ITGA4
Pathway in cancer	hsa-miR-22-5p & MMP1
	hsa-miR-4292 & HRAS
	hsa-miR-136-5p & RAC1

	hsa-miR-1470 & VEGFA
	hsa-miR-125b-5p & VEGFA
Tight junction	hsa-miR-596 & CLDN4
	hsa-miR-3620-3p & CLDN4
	hsa-miR-4292 & CLDN4
	hsa-miR-624-5p & CLDN1
	hsa-miR-596 & CLDN7
	hsa-miR-3620-3p & CLDN7
Cell cycle	hsa-miR-628-3p & CDK1
	hsa-miR-22-5p & CDK1

iv) Potential ceRNAs were predicted for each miRNA-mRNA pair using their MuTaMe Score [Tay et al. Cell. 2011,147(2):344-57], with the cut-off set as MuTaMe Score > 0.005. All 4421 differentially expressed lncRNAs and 3369 differentially expressed mRNAs were entered into this ceRNA prediction as candidates (Supplementary Table 6-7). The interactions were graphed using the Cytoscape software (v2.8.1) (Fig.1b).

6. Figure 1e- the loading control in unbalanced.

Thank you for pointing out this problem. We have performed the western blotting again for detecting the expression of CLDN4 after transfection of miR-596, miR-3620-3p, and miR-4292. The new trial again demonstrated the lower expression of CLDN4 in the miRNA transfected group but had a much better balanced loading control (See Figure below). The new result is shown in Fig.1e.

Figure legend: GC cell line SGC-7901 was transfected with the mimics of miR-596, miR-3620-3p, miR-4292, and negative control. Reduced CLDN4 expression was shown by western blotting analysis and normalized to β-tubulin.

7. To test the biological function of CLDN4 and further verify its association with miR-596 and miR-3620-3p, the authors stably overexpressed CLDN4 in SGC-7901 cells. The gastric cancer cell line (SGC-7901) was established in our laboratory from the surgically resected metastatic lymph node of a female patient. it would be necessary to demonstrate comparable results in other GC cell lines. The reduction of

proliferative capacity and invasion ability caused by knocking down CLDN4 could be largely rescued by inhibition of miR-596 or miR-3620-3p. Variants of miR-596 and miR-3620-3p would be valuable to show that the effect of miRNAs on CLDN4 and cell proliferation is specific.

We agree with the reviewer that it is necessary to demonstrate comparable results in other GC cell lines. The biological function of CLDN4 and its association with miR-596 and miR-3620-3p are now further verified in AGS and HGC-27 cells. We replicated the knockdown experiments, lncRNA experiments, and in vivo experiments with these new cell lines. The experiments in the new cell lines all showed similar results as before. Furthermore, we utilized variants of miR-596 and miR-3620-3p in CLDN4 overexpression experiments. The variants could not inhibit the expression or functions of CLDN4, which verified that the effects of the miRNAs seen in the cell proliferation and invasion assays were specific. We added all of these new results in Fig.2a-2f and Supplementary Fig.2b-2c.

8. In order to further investigate the role of CLDN4 in human GC, we performed deep validations in 62 pairs of GC tissues and NATs. Details on the patient demographics and treatment histories are lacking. With 82.26% of patients showing higher expression in cancer tissue, the transcript level of CLDN4 expression was significant higher in the GC tissues compared with matched NATs. The median survival was 30 months (range, 5-81) in the higher CLDN4 group and 43 months (range, 2-81) in the lower CLDN4 group. The relative expression level of miR-596 and miR-3620-3p in human GC tissues compared with their matched NATs. The correlation between CLDN4 mRNA level and miR-596 or miR-3620-3p transcriptional levels were measured in the same set of patients by Spearman correlation analysis. In Figure 3, it would be valuable to graph panels e, g, and h together to determine whether CLDN4 expression correlates with miR expression of each patient.

Thank you for your suggestion. According to your suggestion, the detailed information on the patient demographics and treatment histories were added in Supplementary Table 12, and we provided the expression of CLDN4, miR-3620-3p and miR-596 for each patient (Supplementary Fig.5).

9. in Figure 4, direct binding experiments are performed to demonstrate lncRNA crosstalk with miRNAs. These experiments should first be performed in multiple different GC cell lines. Second, they need to be performed using miR and lnc mutant variants of the endogenous miR and lnc to demonstrate specificity.

We agree with the reviewer that it is necessary to perform direct binding experiments in different gastric cancer cell lines. According to your suggestion, we performed additional RIP experiments and RNA pull-down in the AGS cell line. Furthermore, we constructed plasmids that could produce lncRNA mutant variants for RIP and RNA pull-down experiments. The results of the RIP experiments showed that the lncRNA-KRTAP5-AS1 RIP was significantly enriched for both miR-596 and miR-3620-3p compared to the RIP of the lncRNA-KRTAP5-AS1 mutant or empty vector (MS2). Likewise, lncRNA-TUBB2A could enrich for miR-3620-3p, but the lncRNA-TUBB2A mutant could not (See Fig.a below). Similar results were observed in the RNA pull-down. MiR-3620-3p could be pulled down by biotin-labeled lncRNA-TUBB2A or lncRNA-KRTAP5-AS1, but it could not be pulled down by the mutant variants of the two lncRNAs. Likewise, biotin-labeled lncRNA-KRTAP5-AS1 could bind miR-596 but the mutant variant could not (See Fig.b below). We added all of these new results in Fig.4d-4e.

Figure legend: (a) The schematic diagram and real-time PCR results of the MS2-RIP method used to identify the binding between lncRNAs and miRNAs in both SGC-7901 and AGS cells. (b) The schematic diagram of the RNA pull down method used to identify the binding between lncRNAs and miRNAs in both SGC-7901 and AGS cells. GC cell lysates of were incubated with biotin-labeled lncRNA-KRTAP5-AS1, lncRNA-TUBB2A, lncRNA-KRTAP5-AS1-mut, and

*lncRNA-TUBB2A-mut. MiRNA real-time PCR was performed after pull down process. Data are shown as mean±SD, n=3. *represents P<0.05, **represents P<0.01.*

10. To evaluate the biological functions of these genes in vivo, different SGC-7901 cells were subcutaneously injected into nude mice. The authors report that CLDN4 promotes SGC-7901 growth in orthotopic transplants. The authors that report that the effect of CLDN4 can be inhibited by miR-596, miR-3620-3p and enhanced by lncRNA-KRTAP5-AS1, lncRNA-TUBB2A. however, these effects are on the inhibition of tumor growth. The authors should demonstrate that, after tumors are allowed to form, that expression of CLDN4, miR-596 and miR-3620-3p modulate tumor growth.

We agree with the reviewer that it would be an important supplement to our data to observe the influence of CLDN4 and the miRNAs and lncRNAs after formation of the tumor. In response to the reviewer's suggestion, we undertook the experiment in vivo. We subcutaneously injected the same amount of tumor cells into the both side of nude mice. After the tumors formed, we injected plasmids of CLDN4, lncRNAs, and agomirs of miRNAs into the tumors of one side and injected their negative controls into the other side. The injections were performed five times with an interval of two days between each injection. Tumor formation in each mouse was monitored every three days before the injection of plasmids and agomirs by taking two-dimensional measurements of the tumor. Three days after the last injection, the mice were sacrificed and the volume and weight of the tumors from both sides were measured. The results showed that CLDN4, lncRNA-TUBB2A, and lncRNA-KRTAP5-AS1 could promote proliferation compared with the negative control even after the tumor formed. Meanwhile, the proliferation of tumors could be suppressed by miR-596 and miR-3620-3p. The results above have been added in Fig.8 as shown below and we have also added notes to the Methods and Results sections detailing the above methods and results in the revised manuscript.

Figure legend: SGC-7901 cells were subcutaneously injected at both right and left armpit region of nude mice. After tumor formation, plasmids of CLDN4, lncRNAs and agomirs of miRNAs were injected into tumors of right side and their negative controls into left side. (a) Treatment of CLDN4 promoted xenograft tumor growth. (b and c) Treatment of miR-596 and miR-3620-3p suppressed xenograft tumor growth. (d and e) Treatment of pcDNA3.1-KRTAP5-AS1 and pcDNA3.1-TUBB2A promoted xenograft tumor growth. Data are mean \pm SD, *means $P < 0.05$, **means $P < 0.01$.

11. The discussion and model require further support from the experiments suggested above.

We appreciate this kind suggestion very much. According to your suggestion, we have added many experiments above and have rewritten the Discussion.

Response to Reviewer #3

1. The bioinformatic (GO and pathway analyses) analyses that led to the focus on CLDN4 is poorly described and based on very small clinical sample numbers. Accordingly, it is not clear to me how the authors ended up analyzing CLDN4.

We appreciate this kind suggestion very much. The process of the bioinformatic analyses that led to the focus on CLDN4 went as listed:

i) Six pairs of GC tissues and non-tumorous adjacent tissues were analyzed via microarray using the Human LncRNA + mRNA Array v2.0 together with the miRCURY LNATM microRNA Array.

ii) Using a Student's t-test, we identified 4421 differentially expressed lncRNAs and 3369 differentially expressed mRNAs which were up-regulated or down-regulated more than 2 fold (Supplementary Table 6-7).

iii) Based on the KEGG database, we performed pathway analysis for differentially expressed mRNAs. We chose the cut-off based on p-val and all pathways with a p-val less than 0.05 proceeded with a deeper analysis. We identified 22 pathways based on the up-regulated genes and 39 pathways based on the down-regulated genes.

iv) Considering the p-val and our previous work, the 61 pathways were filtered down to four pathways of interest. Cell adhesion, pathway in cancer, and cell cycle were among the top candidates and all related to cancer development. Moreover, we selected the tight junction pathway because our previous work demonstrated aberrant expression of CLDN4, a component of the tight junction pathway, in gastric cancer and precursor lesions [Zhu et al. World J Surg Oncol. 2013,11(1):150]. Also, using meta-analysis, we previously found that CLDN4 expression was associated with increasing pT category, tumor size, and lymph node metastasis in patients with gastric cancer [Chen et al. Onco Targets Ther. 2016,9:3205-12]. Therefore, we focused on the tight junction pathway and chose CLDN4 for our key target.

In consideration of the small sample size leading to less stability and accuracy in microarray detection, we measured CLDN4 expression in 104 GC patients to confirm the microarray results. These results further confirmed that CLDN4 expression is significantly higher in GC tissues compared with matched non-tumorous adjacent

tissues ($P < 0.001$, See Fig.3e, Supplementary Fig.4e). Therefore, we ended up analyzing CLDN4. We have added the above content to the Introduction and Supplementary Methods.

2. the data supporting a clinical relevance of CLDN4 is weak. Based on analysis of 62 GC patients the authors state that the CLDN4 upregulation is associated with poor prognosis after gastrectomy. However the authors do not describe how they set the threshold for dividing the CLDN4 expression into low and high. The observed survival difference is very small and likely highly dependent on the defined threshold. “Is the expression levels of the miRs also prognostic? could it be them (independently of CLDN4 that are cause of any survival difference. The authors do not investigate if the CLDN4 observation could be confounded. Is the effect of CLDN4 independent of tumor stage (and other clinical variables)?” To increase confidence that CLDN4 expression is associated with prognosis. The finding should be validated in an independent clinical cohort. Using the same cut-off used here.

We appreciate your questions. Among the 62 GC patients, we used the median expression of CLDN4 (2.343 fold) as the cut-off value to divide the patients into a relatively high group and a relatively low group. In order to confirm this result, we measured the mRNA expression of CLDN4 in an additional independent cohort of 42 GC patients. We used the same cut-off value and again found that patients with higher CLDN4 expression were significantly associated with poorer prognosis ($P = 0.011$, See Fig.a below).

On the other hand, we performed a Cox multivariate analysis among the original 62 GC patients as well as the additional cohort of 42 patients, and the result revealed that higher CLDN4 expression was an independent prognostic factor for poor prognosis in GC ($HR = 2.634$, $95\% CI = 1.536-4.517$, $P < 0.001$, Supplementary Table 10). In addition, we used the median expression of each miRNA (0.509 fold for miR-596; 0.483 fold for miR-3620-3p) as cut-off values to divide the patients into a relatively high group and a relatively low group for each miRNA, respectively. We did not find any significant association between prognosis and the expression of these two miRNAs using Kaplan-Meier analyses ($P > 0.05$ for both, See Fig.b-c below). There

are many factors which might participate in the regulation of CLDN4. Non-coding RNAs may be just one of many. Though miR-596 and miR-3620-3p were not prognostic, these two miRNAs could bind to CLDN4 directly and indeed regulate the expression of CLDN4. Therefore, these miRNAs may affect the prognosis of GC patients indirectly.

Figure legend: (a) Kaplan-Meier analysis of the correlation between overall survival and CLDN4 expression level in the additional cohort of 42 patients. (b and c) Kaplan-Meier analysis of the correlation between overall survival and miR-596, miR-3620-3p expression level in all 104 patients.

3. The authors state that CLDN4 and miR-596, miR-3620-3p are inversely expressed in GC tissues. However, the data supporting this statement are not very solid. It is not surprising that correlation analysis of an upregulated and a downregulated transcript may show some correlation, but this does not mean causality. There may very well be no direct interaction between the two in tissues. The authors will most likely find that the miRs show a weak negative correlation to the majority of the genes they find upregulated. This will in most cases be due to non-causal coincidence. The authors report correlation coefficients of 0.323 and 0.356, indicating that at best 10-12% of the variation in CLDN4 can be explained by the miRNAs. This practically nothing. I wonder if the miRNAs show any correlation to CLDN4 when the tumors and NATs are analyzed separately? This will eliminate the bias that they are comparing opposite deregulated. If CLDN4 and the miRs truly interact they should be negatively correlated also when tumor and NATs are analyzed separately. I wonder if there is matched mRNA and miRNA public available on clinical GC samples that can be used to validate the findings...maybe from the TCGA ???

Following your suggestion, we measured the expression of CLDN4 and miRNAs in the

TCGA database. We found that the RPKM of CLDN4 was significantly higher in 29 GC tissues compared with their matched non-tumorous adjacent tissues ($P < 0.001$, See Fig.a below). This result was in accordance with our finding. However, only five pairs of available data could be used for calculating the expression of miR-3620-3p and no pairs were found that could be used for miR-596. Thus, due to the small sample size, we could not analyze the expression of miRNAs in the TCGA database.

In order to further verify the correlation between CLDN4 and miRNAs, we measured the expression of CLDN4 and miRNAs in an additional independent cohort of 42 GC patients. The result also confirmed that CLDN4 expression was up-regulated in GC tissues and negatively associated with expression of miR-596 ($r = -0.530$, $P < 0.001$) and miR-3620-3p ($r = -0.543$, $P < 0.001$, See Fig.b-c below).

We also explored the correlation between CLDN4 and miRNAs in all 104 GC patients (including the original 62 patients and the additional cohort of 42 patients). Spearman correlation analyses showed negative correlations between CLDN4 expression and miR-596, miR-3620-3p expression ($r = -0.420$ for miR-596, $r = -0.442$ for miR-3620-3p, $P < 0.001$ for both, Fig.3i-3j). These correlation coefficients were larger than before. We added the contents above in Results.

Furthermore, we analyzed the correlations between CLDN4 and miRNAs in tumor and non-tumorous adjacent tissues separately. The results indicated that CLDN4 expression was negatively correlated with the expression of miR-3620-3p ($r = -0.234$ and $P = 0.017$ in tumor tissues, $r = -0.234$ and $P = 0.038$ in non-tumorous adjacent tissues, See Fig.d-e below), but we did not find any association between CLDN4 and miR-596 expression ($r = 0.069$ and $P = 0.489$ in tumor tissues, $r = 0.096$ and $P = 0.331$ in non-tumorous adjacent tissues). Due to the great heterogeneity and individual difference of GC patients, most studies about GC use the tumor and their match non-tumorous adjacent tissues to measure gene expression, and the normalized results may be more reliable. In addition, our results suggest that miR-3620-3p may play a more important role in the regulation of CLDN4 compared with miR-596.

4. That the in vitro and in vivo results are based on two cell lines only. CLDN4 is over-expressed in SGC-7901 cells and knock down is performed in AGS cells. Can the findings be replicated in other GC cell lines or are they peculiarities of the selected lines? The xenograft experiments were also made with the sample cell lines. Without replication in additional cell lines it is impossible to assess if the findings have any general value. The authors should examine a range of GC cell lines to document that the findings are general.

Thank you for pointing this out. We have replicated the overexpression and knockdown experiments for CLDN4 and the lncRNAs in SGC-7901, AGS, and HGC-27 cell lines (Fig.5 and Supplementary Fig.8) The new experiments in each new cell line showed similar results as before: that CLDN4, lncRNA-TUBB2A, and lncRNA-KRTAP5-AS1 could promote GC proliferation and invasion. Moreover, these effects could be inhibited by miR-596 and miR-3620-3p.

Thank you for pointing this out. We have replicated the overexpression and knockdown experiments for CLDN4 and the lncRNAs in SGC-7901, AGS, and HGC-27 cell lines (Fig.5 and Supplementary Fig.8) The new experiments in each new cell line showed similar results as before: that CLDN4, lncRNA-TUBB2A, and lncRNA-KRTAP5-AS1 could promote GC proliferation and invasion. Moreover, these effects could be inhibited by miR-596 and miR-3620-3p.

The xenograft experiments were also replicated using the HGC-27 cell line. In tumor metastasis models, new PET scanning results also support our previous conclusion that CLDN4 promotes metastasis in vivo. Furthermore, the effects of CLDN4 can be inhibited by miR-596 and miR-3620-3p as well as enhanced by lncRNA-KRTAP5-AS1 and lncRNA-TUBB2A (Fig.9f-9g). For the subcutaneous injection model, the results further verified that the effect of CLDN4 on accelerating tumor proliferation could be rescued by miR-596 and miR-3620-3p and exaggerated when lncRNA-KRTAP5-AS1 or lncRNA-TUBB2A were overexpressed (Supplementary Fig.9).

5. There appears to be no analysis of the expression of the lncRNAs KRTAP5-AS1 and TUBB2A in the 62 clinical samples. How are their expression levels correlated with CLDN4 in the clinical samples?

Thank you for raising this question. Real-time PCR showed no significant difference in the expression of the two lncRNAs between GC tissues and matched non-tumorous adjacent tissues ($P=0.306$ for lncRNA-TUBB2A and $P=0.118$ for lncRNA-KRTAP5-AS1). The result of Spearman correlation analysis showed that there was a positive correlation between CLDN4 and lncRNA-TUBB2A expression ($r=0.269$, $P=0.035$). We did not find any correlation between CLDN4 and lncRNA-KRTAP5-AS1 ($r=-0.086$, $P=0.505$). There are many factors which might participate in the regulation of CLDN4, non-coding RNAs may be just one of many. In addition, our results suggest that lncRNA-TUBB2A may play a more important role in the regulation of CLDN4 compared with lncRNA-KRTAP5-AS1.

6. There are no experiments showing which cells actually express CLDN4, the miRNAs and the lncRNAs in the clinical samples!!! The literature contains numerous examples that miRNAs apparently deregulated in epithelial tumors, where in fact expressed by stromal cells and not epithelial cells, and the deregulation was merely a consequence of altered tissue composition. In these situations it does not make sense to overexpress the miRNAs in epithelial cells. The authors should investigate which cell types in the clinical samples do express the essential genes of the CLDN4 network.

Thank you for pointing this issue out. In response to your suggestion, we undertook *in situ* hybridization (ISH) experiments on 20 pairs of formalin-fixed paraffin embedded tissue samples to examine the expression of *CLDN4*, the miRNAs, and the lncRNAs. We found that the miRNAs, lncRNAs, and *CLDN4* were all expressed by epithelial cells instead of inflammatory or stromal cells in the clinical samples. Moreover, this experiment also brought to light that miR-596, miR-3620-3p, lncRNA-KRTAP5-AS1, lncRNA-RUBB2A, and *CLDN4* staining were all mainly localized in the cytoplasm. These results are displayed in Fig.3k and shown below.

Figure legend: Representative images of *CLDN4*, miR-596, miR-3620-3p, lncRNA-KRTAP5-AS1 and lncRNA-TUBB2A expression from GC tissues and non-tumorous adjacent tissues by ISH assays. Original magnification $\times 200$. Scale bars = 100 μm .

Minor comments

1. Many of the figures were small to be properly assessed Fig 1A, Fig 2, Fig 4 and Fig 5.

We have provided larger and clearer figures in the revised manuscript.

2. There is no clinical info provided on the 62 GC cases, only on the 6 used for microarray analysis.

We have added clinical information of all the GC cases in Supplementary Table 12.

3. The P*Si*Check2-CLDN4 vector is not mentioned in the mat and met section. There is no description of what part of CLDN4 that was cloned into the vector making it impossible to assess the validity of the luc-results in fig 1. While the P*si*check2-TUBB2A and P*si*check2-KRTAP5-AS1 vectors are mentioned, there is again no description of what was cloned.

Thank you for this suggestion. Luciferase reporters were generated based on the psiCHECK2 vector (Promega). To construct psiCHECK2-CLDN4, the complete 3'UTR of human CLDN4 mRNA (853nt, UCSC accession no. uc003tzh.2) containing the putative miR-596 and miR-3620-3p binding sites was amplified and cloned into the psiCHECK2 vector. For the lncRNAs, we PCR amplified the full-length sequence of lncRNA-TUBB2A (1052nt, UCSC accession no. uc011dhu.1) and lncRNA-KRTAP5-AS1 (2554nt, UCSC accession no. uc001ltt.1) and cloned them into the psiCHECK2 vector. We have added the details mentioned above to Methods.

4. the mRNA-ceRNA analysis is not described in any detail anywhere

We have added the process of the mRNA-ceRNA analysis to the Results and Supplementary Methods.

5. How was the transcripts shown in figure 1A selected? The figure appears to not contain all deregulated transcripts ? According to page 5 there were several thousand deregulated mRNAs and lncRNAs and a few hundred deregulated miRNAs

We appreciate this kind suggestion very much. Indeed, there were several thousand deregulated mRNAs and lncRNAs and a few hundred deregulated miRNAs, however, we could not display all of them due to space constraints. To present more information, we revised Figure 1A. In the new figure, we present the top 30

up-regulated mRNAs and the top 30 down-regulated mRNAs, as well as the top 30 up-regulated lncRNAs and the top 30 down-regulated lncRNAs. The miRNAs shown in the figure were all differentially expressed between GC tissues and non-tumorous adjacent tissues (greater than 2.0-fold; $P < 0.05$). The new Fig. 1a will appear as:

6. The language is poor many places throughout the manuscript, though particularly in the discussion. Here are a number of examples

Thank you for pointing this out. We have collaborated with a native English speaking biologist to correct all the grammatical errors and edit the language of the whole manuscript.

From page 9 “With 82.26% of patients showing higher expression in cancer tissue, the transcript level of CLDN4 expression was significant higher in the GC tissues compared with matched NATs via Wilcoxon's...”

We changed it to “With 76.92% of patients showing higher expression in cancer tissue, the transcript level of CLDN4 was significantly higher in the GC tissues compared with matched non-tumorous adjacent tissues via Wilcoxon's...”

The following examples are all from the discussion

“Although there is still some controversy on the role of CLDN4 on carcinogenesis 45, many studies have revealed CLDN4’s oncogenic functions such as enhancing proliferation, invasion and EMT”

We changed it to “Although there is still some controversy regarding the role of CLDN4 in carcinogenesis, many studies have revealed oncogenic functions for CLDN4 such as enhancing proliferation, invasion, and EMT.”

“The oncogenic role of CLDN4 in enhancing invasion and EMT can be explained by that claudin family members could activate the MMP family members and ZEB family members, as showed by previous studies”

We changed it to “The oncogenic role of CLDN4 in enhancing invasion and EMT is supported by previous studies showing that claudin family members can activate MMP and ZEB family members.”

“These strengthening effects of CLDN4 are weakened after the exogenous introduction of miR-596 and miR-3620-3p.”

We changed it to “These effects of CLDN4 are decreased upon exogenous introduction of miR-596 and miR-3620-3p.”

“As it is well known that a single miRNA can bind multiple targeting mRNAs through miRNA response elements (MREs)”

We changed it to “As it is well known that a single miRNA can bind multiple target mRNAs through miRNA response elements (MREs).”

“This can be explained that lncRNAs, which possess the capacity to function as ceRNAs, can carry target MREs for various mRNA-targeting miRNAs, resembling the fact that miRNAs can target multiple mRNAs”

We changed it to “These multiplex results are supported by studies showing that lncRNAs, which possess the capacity to function as ceRNAs, can target MREs for various mRNA-targeting miRNAs, similar to how miRNAs can target multiple mRNAs.”

Reviewers' comments:

Reviewer #1 (Remarks to the Author):

The revised manuscript is now much improved and the authors have attempted to address all the concerns raised by the reviewers. The authors have extended their observations to additional gastric cancer cell lines. I have a few comments on the revised manuscript.

There is a contradictory interpretation from the data in Fig.2g-l and Suppl Fig3c-3d. In Fig 2g-l, the authors have downregulated CLDN4 using shRNA followed by miRNA inhibition using miR inhibitors, which results in recapitulation of CLDN4 attenuated proliferation. These results give an impression that miRNAs are playing important role in the absence of CLDN4. However, by combining the data from Fig.2g-l and Suppl Fig3c-3d, the authors conclude that the effects on invasion and proliferation are mainly due to the direct interaction of these two miRNAs with CLDN4. If miRNA-CLDN4 interaction is crucial for this proliferative and invasive phenotype, inhibition of miRNA in the CLDN4 knockdown cells should not have any phenotype. Thus my concern on the specificity of these interactions on the functional outcome of CLDN4 function in gastric cancer cell lines still remains.

The newly added RNA-seq data does not have statistical significance (p-values and FDR). The authors have provided differential expression in FPKM. The authors have not provided how the RNA-seq data was processed and tools used in the analyses.

Figure supplementary 4e shows opposite pattern of CLDN4 expression to what is written in the text: It is more expressed in adjacent tissue compared to the GC tissues.

Minor comments

I am confused with the following sentence, which is in response to reviewers' comments

In consideration of the small sample size leading to less stability and accuracy in microarray detection, we measured CLDN4 expression in 104 GC patients to confirm the microarray results.

Since the entire network analysis is based on the micro-array data, does the above comment has any bearing in the outcome of microarray data?

Please check the sentence "psiCHECK2-lncRNA-KRTAP5-AS1 into GC cells to test the effects the miRNAs".

Comments concerning the response to Reviewer #2:

The authors have responded to all 11 criticisms raised by the reviewer 2. Their response is overall satisfactory barring a few issues.

The reviewer also had concern regarding the selection of the pathway and genes within for the current investigation (comment 5). The selected pathway 'Tight Junction' is statistically significant but not present among the top pathways provided in Suppl Fig. 1. The authors have selected CLDN4 gene from Tight Junction pathway because they have previously worked on this gene. However, their work CLDN4-miRNA-lncRNA is convincing.

That said, Reviewer 2 also had concern (comment 7) on the specificity of CLDN4-miRNA interaction in the regulation of cell proliferation and progression. If CLDN4-miRNA interaction is

specific, the KD of the miRNA in the CLDN4 downregulated cells should not rescue the phenotype. Rescuing of proliferation and cell invasion in the CLDN4 KD cells by the miRNA inhibition says that the miRNAs may play an important role than the miRNA-CLDN4 interaction!

Comment 4 (second part): miR-596 and miR-3620 are expressed at very low levels in GC tissues (n=104). However, Fig 3g-3h and suppl Fig.4e show contrasting data.

Reviewer #3 (Remarks to the Author):

The authors have sufficiently addressed my concerns.

Response to Reviewer #1

1. There is a contradictory interpretation from the data in Fig.2g-l and Suppl Fig3c-3d. In Fig 2g-l, the authors have downregulated CLDN4 using shRNA followed by miRNA inhibition using miR inhibitors, which results in recapitulation of CLDN4 attenuated proliferation. These results give an impression that miRNAs are playing important role in the absence of CLDN4. However, by combining the data from Fig.2g-l and Suppl Fig3c-3d, the authors conclude that the effects on invasion and proliferation are mainly due to the direct interaction of these two miRNAs with CLDN4. If miRNA-CLDN4 interaction is crucial for this proliferative and invasive phenotype, inhibition of miRNA in the CLDN4 knockdown cells should not have any phenotype. Thus my concern on the specificity of these interactions on the functional outcome of CLDN4 function in gastric cancer cell lines still remains.

Thank you for your excellent question. In Fig.2g-2l, we demonstrated that the reduction in proliferative capacity and invasion ability caused by knocking down CLDN4 could be largely rescued by inhibition of miR-596 or miR-3620-3p. In Suppl Fig.3c-3d, we demonstrated that the two miRNAs had little effect on cell proliferation

and invasion when CLDN4 was knocked down. In order to explain this phenomenon, we investigated the expression of CLDN4 after transfecting mimics or inhibitors of miR-596/3620-3p in CLDN4 knockdown cells. The results revealed that the expression of CLDN4 could be rescued by miR-596/3620-3p inhibitors, but could not be further downregulated by miR-596/3620-3p mimics (See figure below).

Figure legend: (a and b) Relative expression levels of CLDN4 determined by real-time PCR in CLDN4 knockdown cells treated with mimics and inhibitors of miRNAs. Data are shown as mean \pm SD, n=3. * represents $P < 0.05$.

In CLDN4 knockdown cells, the expression of CLDN4 decreased about 79% (See Suppl Fig.2c). The expression of CLDN4 was not only suppressed by the shRNA, but it also appears to be regulated by endogenous miR-596/3620-3p, with CLDN4 being increased upon inhibition of the endogenous miRNAs. Thus, the proliferation and invasion could be increased when we inhibited the endogenous miR-596/3620-3p. However, the expression of CLDN4 could not be further decreased following transfection with miR-596/3620-3p mimics in CLDN4 knockdown cells, suggesting that saturating levels of the miRNAs are already present to interact with the decreased levels of CLDN4. Therefore, overexpression of the miRNAs in CLDN4 knockdown cells could not present any further phenotypic change.

2. The newly added RNA-seq data does not have statistical significance (p-values and FDR). The authors have provided differential expression in FPKM. The authors have not provided how the RNA-seq data was processed and tools used in the analyses.

Thank you for bringing this to our attention. We have added the p-values and FDR in the Supplementary table 11. Furthermore, we have added details regarding the process of RNA-sequencing and data analysis in the Supplementary Methods as follows:

The process of the RNA-sequencing and data analysis

Total RNA from each sample was quantified using a NanoDrop ND-1000 instrument. 1-2 µg of total RNA was selected for each sample to construct a sequencing library. Total RNA was enriched by oligo(dT) magnetic beads using the NEBNext® Poly (A) mRNA Magnetic Isolation Module. After processing, the RNA was used to construct a sequencing library using the KAPA Stranded RNA-Seq Library Prep Kit (Illumina), which included procedures for RNA fragmentation, random hexamer-primed first strand cDNA synthesis, dUTP-based second strand cDNA synthesis, end-repairing, A-tailing, adaptor ligation, and library PCR amplification. The completed libraries were qualified on an Agilent 2100 Bioanalyzer and quantified by the absolute quantification qPCR method. To sequence the libraries on the Illumina HiSeq 4000 instrument, the barcoded libraries were mixed, denatured to single stranded DNA in NaOH, captured on an Illumina flow cell, amplified in situ, and sequenced for 150 cycles for both ends on the Illumina HiSeq 4000 instrument.

Image analysis and base calling were performed using Solexa pipeline v1.8 (Off-Line Base Caller software, v1.8). Sequence quality was examined using FastQC software¹. The trimmed reads (trimmed 5', 3'-adaptor bases using cutadapt²) were aligned to the reference genome (hg19) using Hisat2 software (v2.0.4)³. The transcript abundances for each sample were estimated with StringTie (v1.2.3)⁴, and the FPKM⁵ value for gene and transcript levels were calculated with R package Ballgown (v2.6.0)^{6,7}. The differential expression analysis of FPKM was based on Significance B using Perseus software (1.5.3.2)⁸⁻¹⁰. The false-discovery rate (FDR)

adjusted *P* values for multiple testing were calculated with the Benjamini-Hochberg method¹¹.

References:

1. *FastQC*. <http://www.bioinformatics.babraham.ac.uk/projects/fastqc/>.
2. Martin, M. *Cutadapt removes adapter sequences from high-throughput sequencing reads*. *Embnet Journal* **17**, 10-12 (2011).
3. Kim, D., Langmead, B. & Salzberg, S.L. *HISAT: a fast spliced aligner with low memory requirements*. *Nat Methods* **12**, 357-60 (2015).
4. Perteau, M. et al. *StringTie enables improved reconstruction of a transcriptome from RNA-seq reads*. *Nat Biotechnol* **33**, 290-5 (2015).
5. Mortazavi, A., Williams, B.A., McCue, K., Schaeffer, L. & Wold, B. *Mapping and quantifying mammalian transcriptomes by RNA-Seq*. *Nat Methods* **5**, 621-8 (2008).
6. Frazee, A.C., Perteau, G. & Jaffe, A.E. *Ballgown bridges the gap between transcriptome assembly and expression analysis*. *Nat Biotechnol* **33**, 243-6 (2015).
7. Fu, J., Frazee, A.C., Collado-Torres, L., Jaffe, A.E., Leek, J.T. *ballgown: Flexible, isoform-level differential expression analysis*. *R package version 2.6.0*. <https://bioconductor.org/packages/release/bioc/html/ballgown.html> (2015).
8. Leek, J.T., Tyanova, S., Temu, T. & Sinitcyn, P. *The Perseus computational platform for comprehensive analysis of (prote)omics data*. *Nat Methods* **13**, 731-40 (2016).
9. Geiger, T., Mann, M., Cox, J., Cox, J. & Mann, M. *1D and 2D annotation enrichment: a statistical method integrating quantitative proteomics with complementary high-throughput data*. *BMC bioinformatics* **13 Suppl 16**, S12 (2012).
10. Cox, J. & Mann, M. *MaxQuant enables high peptide identification rates, individualized p.p.b.-range mass accuracies and proteome-wide protein quantification*. *Nat Biotechnol* **26**, 1367-72 (2008).
11. Benjamini, Y. & Hochberg, Y. *Controlling The False Discovery Rate - A Practical And Powerful Approach To Multiple Testing*. *Journal of the Royal Statistical Society* **57**, 289-300 (1995).

3. Figure supplementary 4e shows opposite pattern of CLDN4 expression to what is written in the text: It is more expressed in adjacent tissue compared to the GC tissues.

This is a good question. Figure Supplementary 4e shows the expression of CLDN4 in gastric cancer tissues and adjacent tissues. In this figure, we used the ΔCT values to represent the expression of CLDN4, with a larger value indicating lower expression. In order to make them easier to understand, we have improved the figure with new label for the Y-axis. The new figure has been added in Supplementary Figure 4 as follows:

Figure legend: The expression levels of CLDN4 in human GC tissues and matched non-tumorous adjacent tissues were analyzed by real-time PCR. The ΔCT values (CLDN4 normalized to GAPDH) were subjected to the Wilcoxon signed-rank test. Larger ΔCT values indicate lower expression.

Minor comments

1. I am confused with the following sentence, which is in response to reviewers' comments. "In consideration of the small sample size leading to less stability and accuracy in microarray detection, we measured CLDN4 expression in 104 GC patients to confirm the microarray results." Since the entire network analysis is based on the micro-array data, does the above comment has any bearing in the outcome of microarray data?

Thank you for your excellent comments. Our comment does not have direct bearing in the outcome of microarray data. We just emphasized that we measured CLDN4 expression in 104 GC patients.

2. Please check the sentence “psiCHECK2-lncRNA-KRTAP5-AS1 into GC cells to test the effects the miRNAs”.

Following your suggestion, we have rewritten the sentence as “We then transfected the luciferase reporter plasmids psiCHECK2-lncRNA-TUBB2A and psiCHECK2-lncRNA-KRTAP5-AS1 into GC cells to test for potential effects that the miRNAs may have on the expression of the lncRNAs.”

Response to Reviewer #2

1. The reviewer also had concern regarding the selection of the pathway and genes within for the current investigation (comment 5). The selected pathway ‘Tight Junction’ is statistically significant but not present among the top pathways provided in Suppl Fig. 1. The authors have selected CLDN4 gene form Tight Junction pathway because they have previously worked on this gene. However, their work CLDN4-miRNA-lncRNA is convincing.

Thank you for your kind evaluation. We have revised Supplementary Figure 1c to include all significantly down-regulated and up-regulated pathways, included ‘Tight Junction’. The revised figure is shown as follows:

Figure legend: (c) The Enrichment Score (-log10 (P value)) values of all significantly down-regulated and up-regulated pathways.

2. That said, Reviewer 2 also had concern (comment 7) on the specificity of CLDN4-miRNA interaction in the regulation of cell proliferation and progression. If CLDN4-miR interaction is specific, the KD of the miRNA in the CLDN4 downregulated cells should not rescue the phenotype. Rescuing of proliferation and cell invasion in the CLDN4 KD cells by the miRNA inhibition says that the miRNAs may play an important role than the miRNA-CLDN4 interaction!

Thank you for your excellent comment. In our present study, we have demonstrated that the reduction in proliferative capacity and invasion ability caused by knocking down CLDN4 could be largely rescued by inhibition of miR-596/3620-3p, but could not be affected by mimics of miRNAs. In order to explain above phenomenon, we investigated the remaining expression of CLDN4 after transfecting mimics or inhibitors of miR-596/3620-3p in CLDN4 knockdown cells. The results revealed that the expression of CLDN4 could be largely rescued by miR-596/3620-3p inhibition, but it could not be further downregulated by miR-596/3620-3p mimics (See figure below).

*Figure legend: (a and b) Relative expression levels of CLDN4 were determined by real-time PCR in CLDN4 knockdown cells treated with mimics and inhibitors of miRNAs. Data are shown as mean \pm SD, n=3. * represents $P < 0.05$.*

In CLDN4 knockdown cells, the expression of CLDN4 decreased about 79% (See Suppl Fig.2c). The expression of CLDN4 was not only suppressed by the shRNA, but

it also appears to be regulated by endogenous miR-596/3620-3p, with CLDN4 being released upon inhibition of the endogenous miRNAs. Thus, the proliferation and invasion could be increased when we inhibited the endogenous miR-596/3620-3p. However, the expression of CLDN4 could not be further decreased following transfection with miR-596/3620-3p mimics in CLDN4 knockdown cells, suggesting saturating levels of the miRNAs are already present to interact with the decreased levels of CLDN4. Therefore, overexpression of the miRNAs in CLDN4 knockdown cells could not present any further phenotypic change.

3. Comment 4 (second part): miR-596 and miR-3620 are expressed at very low levels in GC tissues (n=104). However, Fig 3g-3h and suppl Fig.4e show contrasting data.

This is a good point. The explanation is as follows:

Figure legend: The relative expression levels of miR-596 and miR-3620-3p in human GC tissues compared with their matched non-tumorous adjacent tissues.

In this figure, we showed the relative expression levels of miR-596/3620-3p in each human gastric cancer tissue compared with their matched non-tumorous adjacent tissue, which were represented by $-\Delta\Delta CT$ value. Larger $-\Delta\Delta CT$ values indicated higher expression of the miR-596/3620-3p.

Figure legend: The expression levels of miR-596, miR-3620-3p in human GC tissues and matched non-tumorous adjacent tissues were analyzed by real-time PCR. The Δ CT values (miRNAs normalized to U6 snRNA) were subjected to the Wilcoxon signed-rank test. Larger Δ CT value indicated lower expression.

In this figure, we showed the expression of miR-596/3620-3p separately in gastric cancer tissues and their matched non-tumorous adjacent tissues represented by Δ CT value. Larger Δ CT values indicate lower expression of miR-596/3620-3p. Thus, the results are completely consistent and accurate in both figure sets. In order to make them easier to understand, we have improved the figures with new Y-axis labels.

Thank you again for your kind comments and suggestions.

REVIEWERS' COMMENTS:

Reviewer #1 (Remarks to the Author):

The revised version addresses all my concerns.